# Study on the diffusion and deposition law of pore slurry in gangue filling zone based on CFD-DEM coupling

**Zhongkui Ji[1,2], Lijun Gao[3], Shuquan Guo[3], Kui Sun[1,4], Wanchao Ma[1,4], Boqiang Wu[2]\*, Xiaoyuan Xue[2], Pan Chen[2]**

1 Key Laboratory of Mining Geological Hazards Disaster Mechanism, Prevention and Control, Ministry of Natural Resources, Xi'an, China, 2 China Coal Industry Xi'an Research Institute (Group) Co., Ltd, Xi'an, China, 3 Shaanxi Coal Group Shenmu lemon tower Mining Co., Ltd, Yulin, Shaanxi, China, 4 Shaanxi Provincial Geological and Environmental Supervision and Monitoring General Station, Xi'an, China

\* kknder@vip.qq.com

**Data Availability Statement:** All relevant data are available at Wu, Boqiang (2023), "Slurry velocity and particle velocity", Mendeley Data, V1, doi: 10.17632/6z2wfrw4sm.

## Abstract

In this study, the slurry diffusion in a cavity filled with coal gangue was studied by combining experimental and numerical simulation methods. By calibrating slurry and particle materials, the grouting process in coal gangue filling area is simulated successfully, and the change of slurry diffusion flow field and particle movement and settling process in different dimensions are deeply analyzed. Both experimental and numerical simulation results show that the particle settlement presents a bell-shaped curve, which is of great significance for understanding the particle movement and settlement behavior in the filling cavity. In addition, it is found that the grouting speed has a significant effect on the particle settlement during the slurry diffusion process. When the grouting speed increases from 0.1m /s to 0.2m /s, the particle settlement and diffusion range increases about twice. In the plane flow field, it is observed that the outward diffusion trend and speed of grouting are more obvious. It is worth noting that in the whole process of grouting, it is observed that with the increase of grouting distance and depth, both the velocity of slurry and particles show a trend of rapid initial decline and gradually slow down, and the flow velocity of slurry near the grouting outlet at a flow rate of 0.2m/s is 2–4 times that of 0.1m/s. This provides important enlightenment for the porous seepage effect at different grouting speeds.

## 1. Introduction

Filling and grouting with gangue in goaf is a common mining engineering technology module. By filling the voids of the gangue skeleton with slurry, the resource utilization rate of the mine can be effectively improved, and problems such as ground collapse caused by goafs can be prevented [1, 2]. Filling gangue-filled voids with slurry is an effective way to use resources and deal with mining goafs. Its research significance is very important, mainly in the following aspects: protect the stability of underground mines, reduce the amount of waste rock accumulation, reduce mine environmental pollution, reduce the possibility of geological disasters,

**Funding:** This work is supported by Tiandi Science and Technology Co. Ltd. Science and Technology Innovation Venture Capital Special Project (2023-TD-ZD004-003), Science and Technology Innovation Fund of Xi'an Research Institute of CCTEG (2023XAYJS11).] The funder has no role in research design, data collection and analysis, but provides funding for experiments and the purchase of computing equipment.

**Competing interests:** The authors have declared that no competing interests exist.

reduce the risk of accidents, reduce the impact on the surrounding ecosystem, protect its balance and stability. Therefore, studying the law of gangue filling in goafs with grouting injections has important engineering practice value and scientific significance, which is of great importance to the sustainable development of mining and environmental protection.

Many scholars have conducted in-depth research on the diffusion law of gangue slurry from different perspectives. Their research provides important information about the diffusion law of gangue slurry, revealing the infiltration process and mechanism of the slurry in different media from different angles. Among them, Liu's study found that the filtration effect has a significant impact on the velocity and concentration of gangue slurry, making it difficult to evaluate the diffusion range. He established a theoretical calculation model considering temporal and spatial effects of infiltration and deduced the "water-cement ratio change matrix", revealing the basic mechanism of porous medium filtration [3]. Liang Li proposed a technology for fluidized injection of gangue slurry to fill subsidence areas and studied the distribution law of residual cavities. Understanding the distribution law of remaining space is important for filling effects. He simulated coal gangue slurry diffusion in goaf using COMSOL simulation software and found that "cavity-pore" multi-type residual spaces gradually expand [4]. In addition, researchers have also studied the effects of major fissures in fracture networks and slurry viscosity on diffusion, developed dynamic water-injection grouting simulation experimental devices for fractured rock masses. They determined experimentally and numerically that grouting pressure and time affect rock mass elastic modulus, studying modification effects of high viscosity slurries [5]. Wenzhe Gu analyzed the phase composition and microstructure characteristics of gangue to study its rheological properties affected by concentration, proposing a technique to make gangue into slurry and transport it through pipelines to fill goafs [6]. Huazhe Jiao explored pore structure shear evolution and bottom flow slurry dehydration behavior using high-precision CT scanning and 3D reconstruction techniques, revealing thickening mechanisms [7–10]. Their research results and numerical simulations indicate that shearing destroys pore structures within coagulation networks, fills voids, reduces absolute permeability thereby increasing bed concentration [11]. In addition, C Hou et al. studied the seepage process of bentonite mud injection into soil skeleton and its significant impact on the stability of tunnel faces [12]. Senlin Ling studied the effect of pore characteristics on the permeability performance of grouting materials in PAM, providing theoretical guidance for grouting construction [13]. These studies comprehensively considered experimental and numerical simulation methods, exploring in depth the injection and seepage laws of slurry in pipelines and gangue pores from macro and micro perspectives, providing strong theoretical support for gangue backfill grouting. These studies comprehensively consider the two methods of experiment and numerical simulation, and deeply explore the injection and seepage law of slurry in pipelines and coal gangue pores from macro and micro perspectives, providing a strong theoretical support for coal gangue filling and grouting. Then, in these studies, on the one hand, the slurry is treated as a single-phase fluid, and on the other hand, the pore structure of the gangue particle accumulation is generalized as a pressure drop area. However, the flow and deposition of particles in the porous structure of the slurry are ignored, which leads to insufficient understanding of the flow and deposition law of particles.

In order to further explore the mechanism of slurry flow and particle deposition in the pore structure of coal gangue, the pore skeleton model of coal gangue was established by experiments and numerical simulation. Slurry particles are injected into gangue pores at the entrance of the pipeline, and the seepage and deposition rules of slurry in different pore size structures are deeply analyzed [14]. Since it is important to highlight the deposition law and flow trajectory of particles, CFD-DEM coupling simulation method is adopted to conduct grouting simulation research [15]. The complex flow field and the particle motion behavior can be

simulated, which is the kernel superiority of the combination of fluid mechanics and discrete element simulation [16, 17]. Taking slurry as a continuous fluid with interparticle interaction, the process of seepage and settlement of slurry in the pore structure can be simulated. More importantly, it provides information on particle motion at the microscopic scale, including their interaction phenomena such as sedimentation, blocking, and the selection of accurately described seepage paths.

In this study, the experimental results are compared with the coupled simulation results of CFD-DEM, which shows the validity of the simulated structure. At the same time, by simulating the pore structure of different pore sizes, the seepage settlement behavior of slurry under different grouting flow rates can be studied, and the influence of pore size on slurry diffusion and filling in the gangue structure can be revealed. It is of great significance to optimize grouting process and improve filling effect. The specific contents of this paper are arranged as follows: The second section describes the principle of CFD-DEM coupling method in detail. This method uses DEM technology to establish pore model of coal gangue, and combines CFD technology to simulate multiphase particle flow in grout. The third part introduces the method of parameter calibration and numerical model construction in detail. In addition, the flow experiment of colored particle slurry was carried out to verify the accuracy and reliability of the numerical simulation results. In the fourth part, the numerical simulation results are comprehensively analyzed and discussed, and the law of particle settlement during grouting and slurry flow is mainly studied. Finally, the fifth part summarizes the research results in detail, and provides guidance and suggestions for engineering practice, and further provides guidance for engineering application.

## 2. Numerical method

### 2.1 Construction of DEM coal gangue pore model

The construction of pore structure model of coal gangue by discrete element method (DEM) involves the construction of pore structure model of coal gangue on particle scale. The process of building this model consists of the following steps. The geometric size and parameters of coal gangue particles were collected. According to the experimental analysis results of pore structure, the pores are randomly generated in the particle model or according to the observed pore morphology data. The appropriate boundary conditions are set according to the specific simulation requirements. Through these steps, the pore model of coal gangue based on measured data and characteristics and pore structure, as shown in Fig 1.

### 2.2 Principle of CFD-DEM coupling technolog

CFD-DEM coupling technology combines computational fluid dynamics (CFD) with discrete element method (DEM) to simulate multiphase particle flow and the interaction between fluid and particles. Its core computing modules include fluid dynamics simulation, discrete element method simulation and fluid-particle coupling.

CFD is based on Navier-Stokes equations and discretization, by solving these equations to simulate the motion and flow of fluids. It takes into account fluid continuity, fluid dynamics equations and boundary conditions. The core control equation can be expressed as NS equation coupled with particle motion equation [18].

Considering single-phase flow and ignoring the influence of solid phase on continuous phase flow, continuity and momentum can be written as Eqs (1) and (2) [19].

$$\frac{\partial(\rho \mathbf{u})}{\partial t} + \nabla \cdot (\rho \mathbf{u}\mathbf{u}) = -\nabla p - \nabla \cdot \tau + \rho \mathbf{g}, \tag{1}$$

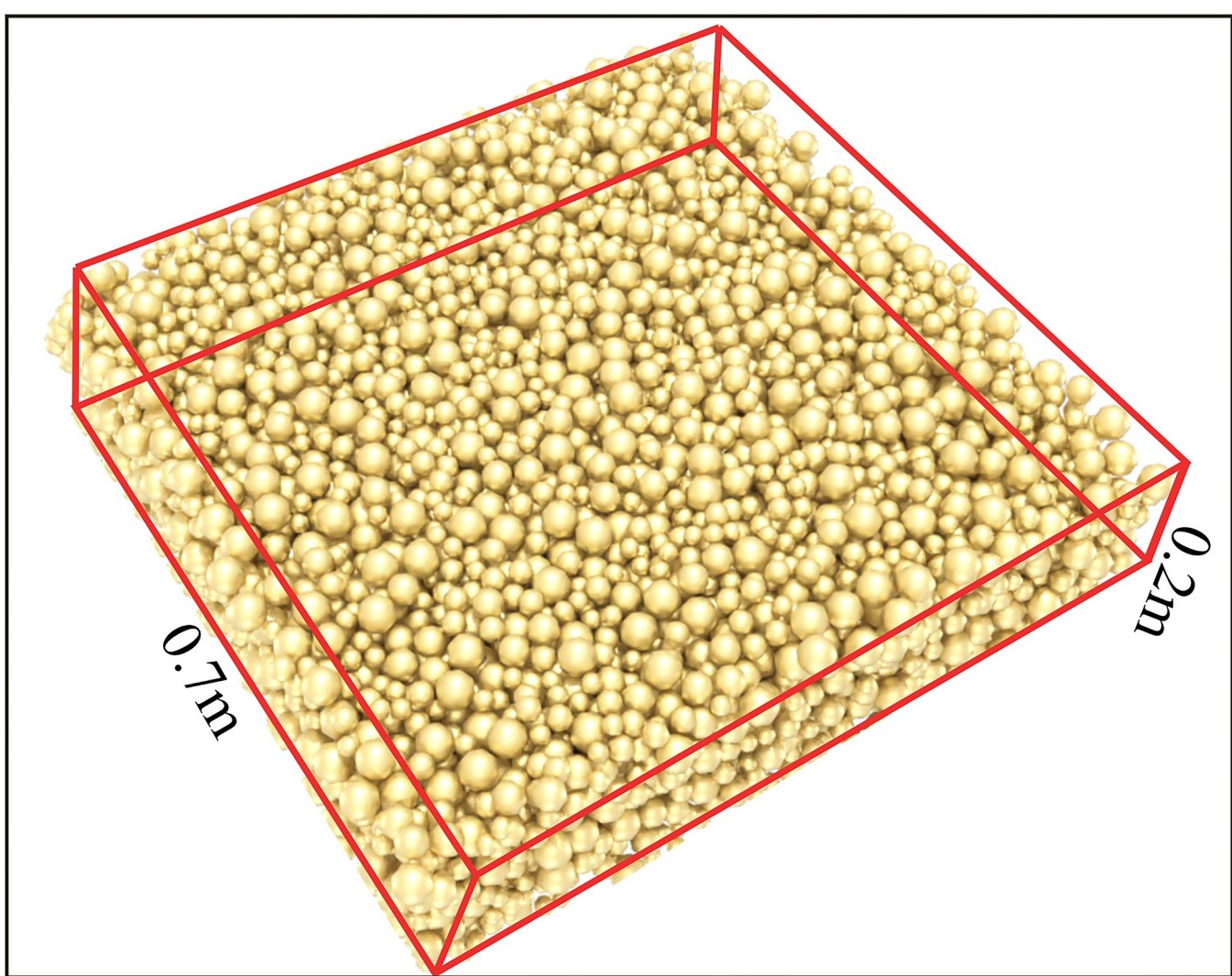

**Fig 1. Discrete element simulation of gangue accumulation zone.**

$$\frac{\partial}{\partial t}(\rho) + \nabla \cdot (\rho \mathbf{u}) = 0, \tag{2}$$

$\nabla$ is the vector of Differential operator, $p$ is the pressure, $\rho$ is the fluid density, $\mathbf{u}$ is the velocity, $\mathbf{g}$ is the Gravitational acceleration, $\tau$viscous shear stress can be defined by Formula (3) [20]。

$$\tau = [\mu(\nabla \cdot \mathbf{u} + \mathbf{u}\Delta)], \tag{3}$$

By substituting the viscous shear $\tau$ in a Newtonian fluid from Eq (3) into Eq (1) and considering an incompressible Newtonian fluid (where the shear deformation of the fluid under stress is linear), Eq (4) can be obtained.

$$\frac{\partial(\rho \mathbf{u})}{\partial t} + \nabla \cdot (\rho \mathbf{uu}) = -\nabla p - \nabla \cdot [\mu(\nabla \cdot \mathbf{u} + \mathbf{u}\Delta)] + \rho \mathbf{g}, \tag{4}$$

As the Reynolds number increases (i.e., the ratio of inertial to viscous forces) the flow state

changes from creep (the convective term in the momentum equation can be ignored), to laminar, to transitional, and finally to turbulent. If the flow state changes, the treatment of the NS equations changes, and the Reynolds number macroscopic scale is defined as Eq (5) [21, 22].

$$\mathrm{Re}_p = \rho_f d_{p,i} \mathbf{v}_p / \mu_f, \tag{5}$$

In Finite volume method (FVM) according to the Reynolds number can be divided into laminar, transitional and turbulent flows. Turbulence models (RANS($k-\varepsilon$、 $k-\omega$)、 LES、 DNS) are required for high Reynolds number calculations [23, 24]. In this study, it can be determined by the flow velocity and density of the fluid that the Reynolds number is 8800 at 0.1m/s, and the entire calculated flow is in a turbulent state, so the turbulence model RANS is adopted to calculate the slurry flow.

Based on Newton's second law, the mechanical interaction between particles is considered, and the motion and collision behavior of particles are simulated. It describes the motion of particles by solving the mechanical equations between particles as discrete units.

In the framework of the DEM, the motion of the particles is solved over time by a large set of coupled ordinary differential equations to obtain the kinetic behavior of the particle flow. The motion of the particles is governed by the Newtonian equations of motion describing the translational and rotational motion of a spherical particle (motion with six degrees of freedom in three dimensions). The particle $i$ in space realizes the contact collision, rotation and other motions in space through the contact with other particles j, the solid wall and its own gravity [25], and the advection control equation of the particles under the force in the DEM can be expressed as Eq (6).

$$m_i \frac{d\mathbf{v}_i}{dt} = m_i \mathbf{g} + \sum (\mathbf{f}_{i,j} + \mathbf{f}_{i,w}), \tag{6}$$

$m_i$ denotes the mass of particle $i$, $\mathbf{v}_i$ denotes the velocity of particle $i$, $\mathbf{f}_{i,j}$, $\mathbf{f}_{i,w}$ is the force generated by particle-particle and particle-wall collisions, $\mathbf{g}$ is the local acceleration of gravity.

On the other hand, the rotational motion of the particles is controlled by Eq (7).

$$\mathbf{I}_i \frac{d\boldsymbol{\omega}_i}{dt} = \sum (\mathbf{T}_r + \mathbf{T}_t), \tag{7}$$

Where, $\mathbf{I}_i$ is the moment of inertia matrix, $\mathbf{T}_t$ is the tangential force $\mathbf{t}_{ij}$ moments generated for other particles or wall surfaces, $\mathbf{T}_r$ is the rolling friction torque, $\boldsymbol{\omega}_i$ is the particle angular velocity.

In CFD-DEM coupling, the influence of particles on fluid is considered by introducing corresponding physical models, including the particle-fluid interaction model and the volumetric force model of particles on fluid. The interaction coupling between fluid and particle is realized through the transfer of physical quantity and force information between CFD and DEM methods.

The governing equation of CFD-DEM coupling can be expressed as the Eq (8).

$$\frac{\partial(\rho\mathbf{u})}{\partial t} + \nabla \cdot (\rho\mathbf{u}\mathbf{u}) = -\nabla p - \nabla \cdot \tau + \rho\mathbf{g} + +\mathbf{f}_{pf}, \tag{8}$$

$\mathbf{f}_{pf}$ is the interaction force between fluid and particles.

In this study, the above governing equations can be described in detail in the coupling flow through Fig 2.

The entire calculation process can be expressed as setting the initial velocity of the fluid in CFD simulation, solving the Navier Stokes equation, and using CFD method to simulate fluid motion. Construct the pore structure of gangue particles in DEM simulation. Coupling CFD

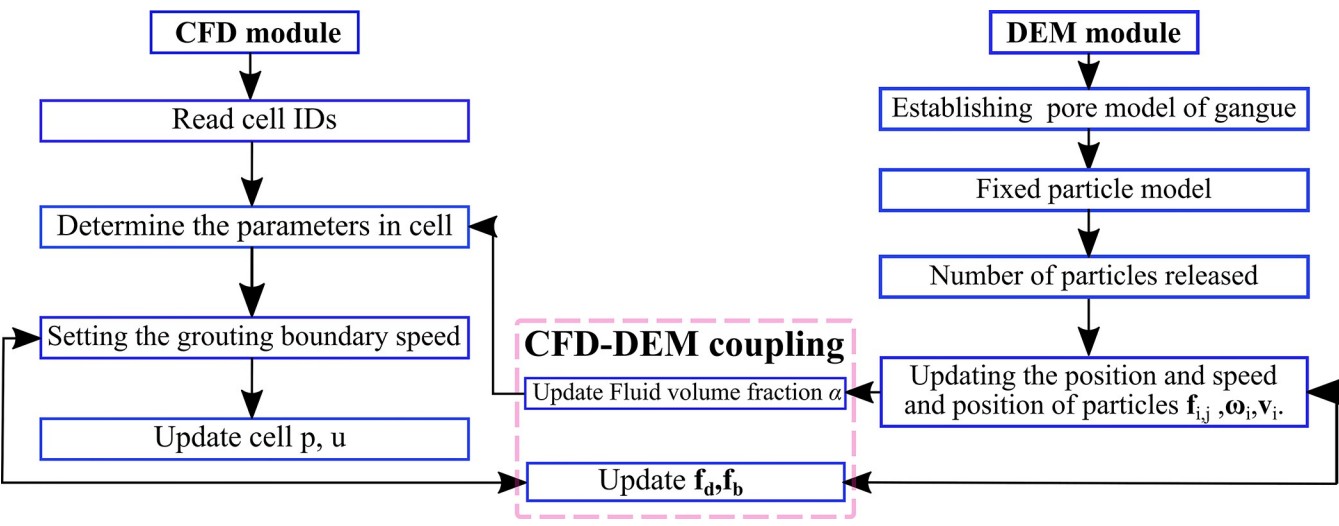

**Fig 2. Coupling flow chart of slurry in gangue flow.**

results with DEM results. At each time step, the drag force received by particles is calculated based on the fluid velocity field to update their motion state. Iterative loop until the predetermined calculation time or convergence standard is reached. By using CFD-DEM coupling technology, the multi-phase particle flow process such as particle suspended slurry can be simulated more accurately. This coupled approach provides an in-depth understanding of complex flow phenomena and is applied in engineering to simulate and optimize particle-related processes.

## 3. Parameter calibration and modeling

It is indeed very important to determine the relevant parameters of slurry particles in numerical simulations, and the determination of particulate parameters by angle of repose tests is a commonly used method [26–29]. The angle of repose test can help to determine the angle formed by the granular material during the stacking process and is further applied to the design of the granular stack and the study of the physical properties of the granular material.

The test steps of the rest Angle test include the selection of granular material, the preparation of the test vessel, the pouring of granular material, the observation of the accumulation pattern (measuring the Angle formed by the pile edge with the appropriate tool (e.g., ruler, Angle gauge, etc.), and data analysis.

The quantitative information of the stacking Angle of granular materials during the stacking process can be obtained through the repose Angle test. The information can be used to set and calibrate the parameters of slurry particles and gangue materials in numerical simulation, so as to improve the accuracy and reliability of numerical simulation. In this study, the angle of rest test can provide physical property information of slurry particles and gangue particles, such as particle restitution and friction coefficient. The density and size of the particles can be obtained in the test, but the Poisson's ratio, friction coefficient and restitution coefficient of the particles are difficult to obtain. Therefore, the resting Angle test is adopted, that is, the packing Angle is measured when the particles are packed, and then the parameters of the particles are changed in the numerical simulation to measure the packing rest Angle. When the packing rest Angle is close to the packing rest Angle, the parameter calibration may be considered complete. The angle of rest test of this study is shown in Fig 3. In Fig 3, the angle of repose α、β was obtained through experiments Fig 3(A) and numerical simulations Fig 3(B).

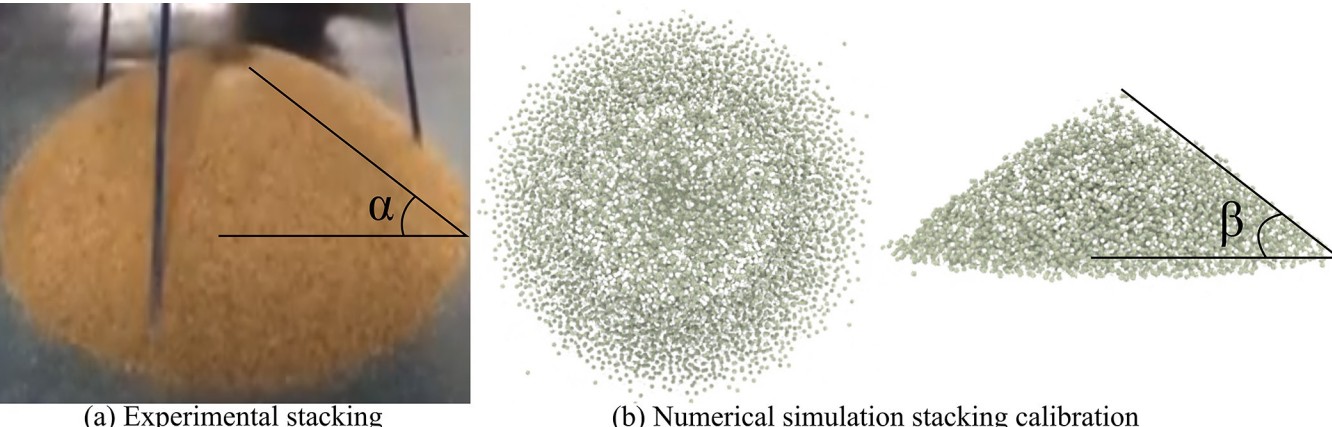

(a) Experimental stacking (b) Numerical simulation stacking calibration

**Fig 3. Measurement of the stacking angle of slurry particles.**

This information also has important reference value for the engineering design and research of granular materials. The final discrete element parameters are shown in Table 1.

The fluid parameters of the slurry mainly include viscosity and density. When the viscosity of the slurry is large, the adhesion between the fluid particles is enhanced, and the fluid flow becomes difficult. The greater the viscosity, the slower the flow. When the viscosity of the slurry is small, the fluidity increases and the fluid flow speed is faster. The slurry with higher density has greater inertia force, which makes the probability of collision of fluid particles increase and the flow becomes more intense. The slurry with less density has less inertia and smoother flow.

The viscosity of the slurry tested in this study is $1.0e^{-5}$ m/s$^2$ and the density is 1100 kg/m$^3$. On the basis of determining the material for numerical simulation, the convergence of the mesh and the relationship between the mesh particles are further analyzed, as shown in the Fig 4.

In Fig 4, when the fluid mesh decreases from 0.02m to 0.01m, the monitoring of the flow velocity at the same point shows no significant change, indicating convergence under the conditions of 0.02-~0.01m. In addition, considering whether the drag model is satisfied between the fluid mesh and particle size, The size of the skeleton particles is close to or even larger than that of the fluid grid, but because the skeleton particles only occupy the volume of the fluid without calculating the drag force, only the relationship between the size of the grout particles and the fluid grid is required. In the numerical simulation, the size of the grout particles is 3 times lower than that of the fluid grid, so it meets the requirements of coupling calculation.

**Table 1. Parameters of waste rock and tailings determined by angle of repose tests.**

| Material parameters | Gangue | Slurry particles |
|---|---|---|
| Density (kg/m$^3$) | 1560 | 1400 |
| Poisson's ratio | 0.35 | 0.28 |
| Young's modulus (Pa) | $5.0*10^9$ | $2.0*10^8$ |
| Dynamic friction coefficient | 0.05 | 0.06 |
| Static friction coefficient | 0.4 | 0.5 |
| Coefficient of restitution | 0.01 | 0.02 |
| Particle size (m) | 0.008~0.015 | 0.001~0.002 |

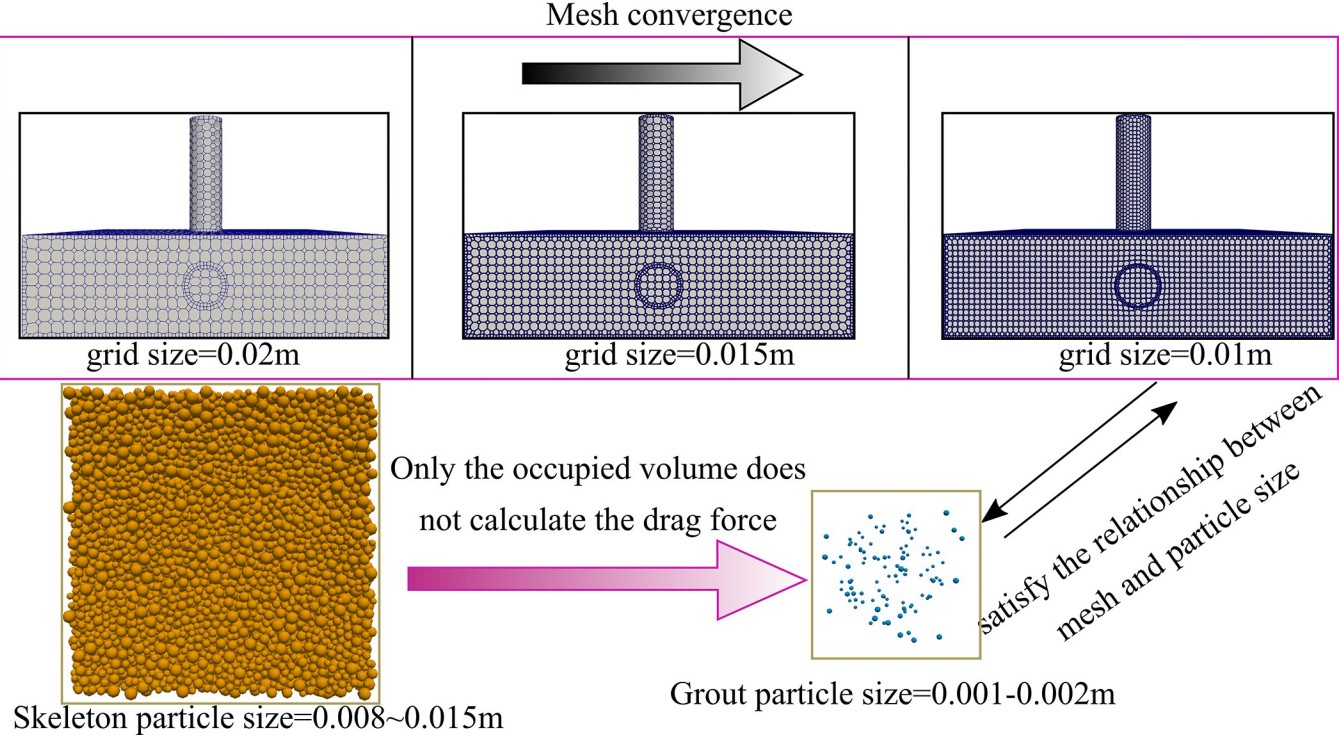

Fig 4. Calculation grid of fluid domain and its relation with particles.

## 4. Analysis and discussion of results

On the basis of determining the parameters, the coupled CFD-DEM solution is realized by setting the fixed boundary conditions of fluid and particle release, and different inlet flow rates are used in this study to understand the effect of injection flow rate (converted to velocity 0.1m/s, 0.2m/s) on the percolation pattern of waste rock and tailings in the waste rock tailings seepage system, and the boundary settings and computational domains of numerical simulation (fluid and solid domains) are shown in Fig 5 shown.

In this section, the changes involved in the numerical simulation process shown in Fig 3 are analyzed and discussed in detail.

### 4.1 Discussion on the law of particle deposition in the process of slurry injection

Firstly, the pore space formed after gangue particles occupied the pore space was observed, so as to further ensure the validity of the results through the comparison of the numerical simulation porosity and the porosity in the test, as shown in Fig 6.

In the numerical simulation, different grouting speeds (0.1m/s, 0.2m/s) were set at the boundary of the grouting pipe, and the motion and deposition laws of the particles were observed to understand the movement of particles in the pore space. The movement and distribution of particles at different grouting moments are very important for the study of particle movement. Therefore, the movement and deposition positions of particles in pores at different grouting moments are monitored. As shown in Fig 7.

Fig 7 shows some key points in the flow of slurry from the pipe into the gangue zone. First, when injecting the slurry, the slurry velocity gradually increases due to gravity. When the

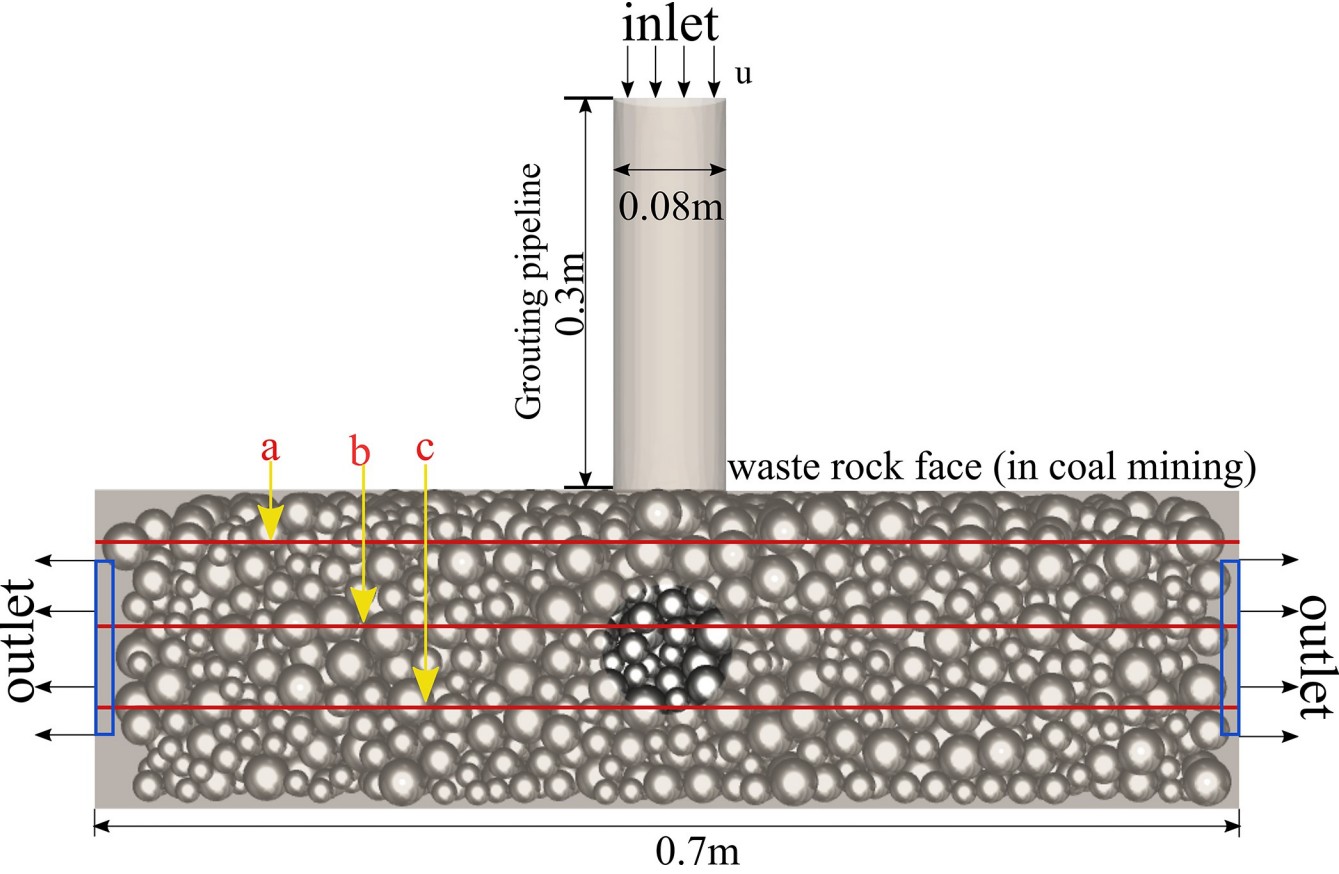

**Fig 5. Schematic dimensioning of the numerical simulation of grouting.**

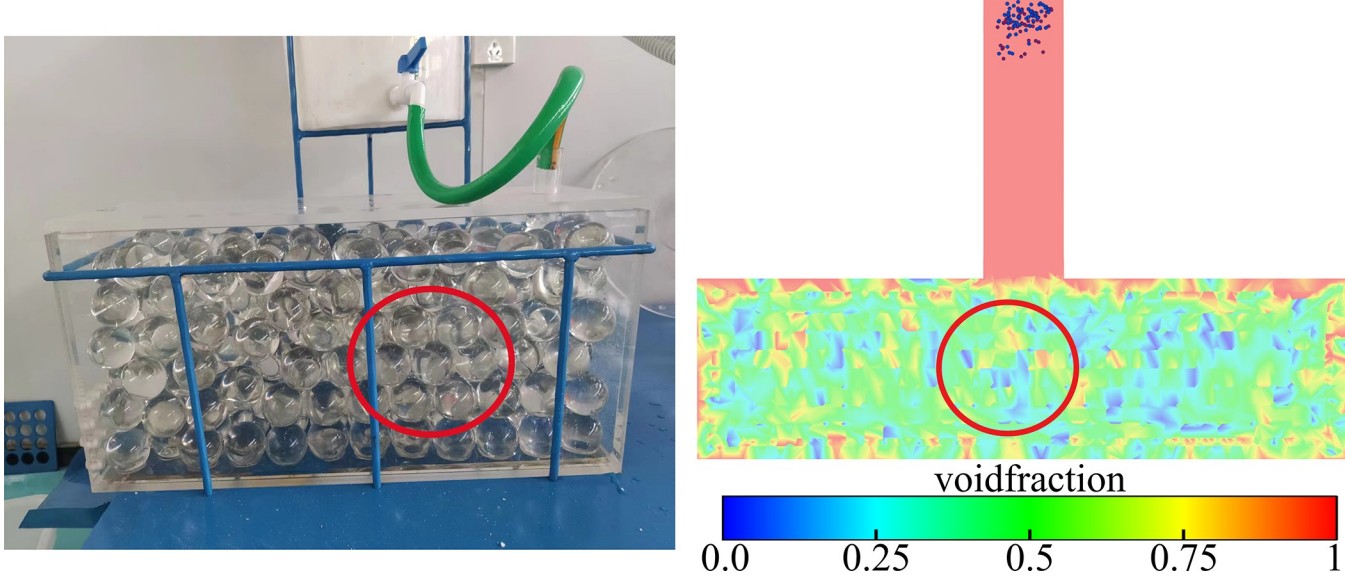

**Fig 6. Fluid can pass through the pore rate in the calculation domain.**

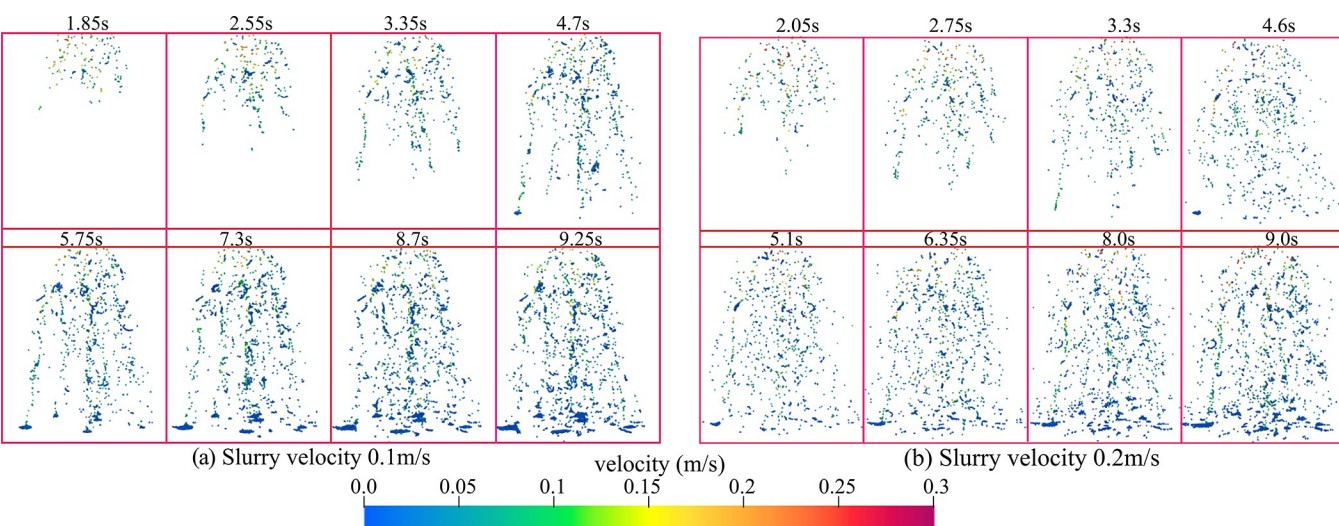

**Fig 7. Slurry particle flow and deposition at different moments.**

slurry contacts the gangue area, it is resisted by the gangue as well as the pores of the gangue structure, which causes the slurry flow to become slow. As the slurry flows into the interior of the gangue structure, particles flow, deposit and spread inside the slurry. The flow pattern of particles in the pore space under different grouting flow rates is also shown in Fig 7. In the pore space, the microparticles show movement along the channels and deposition at the blocked pores. While in the channel with better connectivity, the particles would accumulate at the bottom and then spread around, forming claw or bell-shaped flow trajectories. Meanwhile, by comparing the distribution patterns of slurry particles at different grouting velocities, it can be observed that the distribution width of slurry particles at a flow rate of 0.2 m/s is larger than that at a flow rate of 0.1 m/s. In conclusion, the flow and diffusion process of the slurry when it enters the gangue area from the pipe is shown in Fig 7, and the changes in the distribution and diffusion range of the particles under different grouting velocities are presented.

The diffusion radius of the slurry particles at the bottom can progress to prove the diffusion area under different grouting velocities to further understand the influence of the grouting velocity on the diffusion range of the slurry, as shown in Fig 8.

It can be observed through Fig 8 that the spreading range of the slurry in the XY plane at 0.2m/s grouting speed is significantly larger than that at 0.1m/s speed, in the X direction the range of grouting at 0.1m/s is between about 0.15–0.33m position, while in the case of 0.2m/s the range of grouting becomes about twice as big as that at 0.0–0.35m position. In the Y-direction the distribution range is compared 0.1m/s ranges between 0.07–0.18m position, whereas in the case of 0.2m/s the grouting range becomes about double at 0.02–0.225m. The above results show that increasing the slurry injection speed will make the slurry and particles in the gangue pore space diffusion more obvious particles are also more widely distributed, compared with the test process of the slurry particle distribution can also prove that the numerical simulation results of the shape of the slurry distribution is close to the test, the test results are shown in Fig 9.

The results of the grouting experiments in Fig 9 clearly show that the slurry-particle deposition interface exhibits a bell-shaped half. This is consistent with the whole range of bell shape formed by the slurry particles deposited during numerical simulation [30, 31].

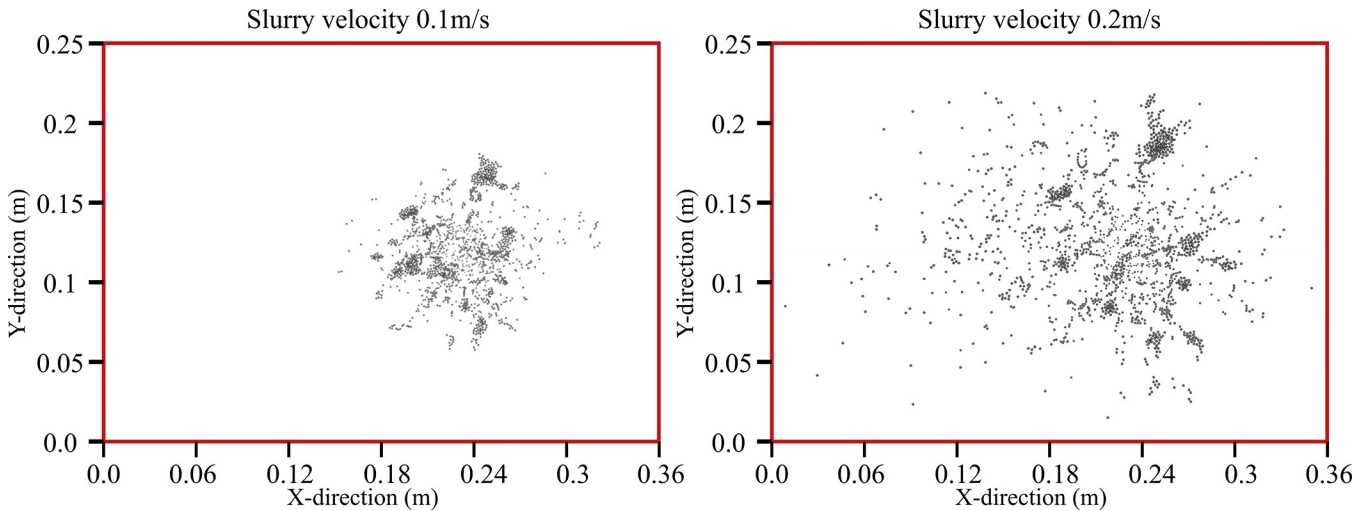

**Fig 8. Range of slurry diffusion.**

## 4.2 Analysis of the regularity of slurry flow

The change of the flow rate of the slurry in the longitudinal direction as well as in the plane is more concerned in the process of grouting, which is a key factor affecting the scope of grouting, and the flow law of the slurry in the longitudinal direction is analyzed, as shown in Fig 10.

Through the grouting pipe in Fig 10 and the gangue filling area of different moments of the velocity change can be seen from the pipeline to the gangue filling area velocity obviously shows the gangue resistance due to the influence of the velocity change, in addition, can also see the gangue internal pore at the change of the velocity maps, and further compared to the vector diagram of the flow field can be a better understanding of the gangue in the pore of the flow, as shown in Fig 11.

Through the change of flow vector within the pore space of gangue pile in Fig 11, it can be observed that the slurry flows along the pore space toward the exit of the surroundings, i.e., from the grouting pipeline to the gangue pile through the internal pore space to the exit of the surroundings, which is the main pathway of the slurry diffusion in the pore space of the gangue. The flow field of the whole profile also shows the accumulation of particles to form a flow field outside the bell shape which is similar to the results of the experimental and numerical simulation of particles, which can be further expressed as the flow field at the accumulation of slurry particles is difficult to form the outward diffusion of the power, and the flow field of the whole profile can be interpreted as shown in Fig 12.

Through the vector schematic in Fig 10, it can be clearly observed that the field in the stacking area mainly presents downward flow rather than diffusion to the surrounding area, so in this part of the area mainly presents particle stacking, the vector flow theory in Fig 11 agrees with the experimental and numerical simulation results, and proves the validity of the numerical and experimental results. On this basis, the flow vectors in the XY plane are further analyzed, as shown in Fig 13.

Through the flow of velocity vector in the figure, it can be seen that the whole slurry flows along the pore space in all directions, and at the flow rate of 0.1m/s the slurry spreads from the position of the center pipe in all directions, and the vector of the whole slurry flow is relatively small, while the density and size of the vector of the slurry flow inside the pore space are obviously larger than that of 0.1m/s at the time of 0.2m/s, which indicates that the slurry flow at the

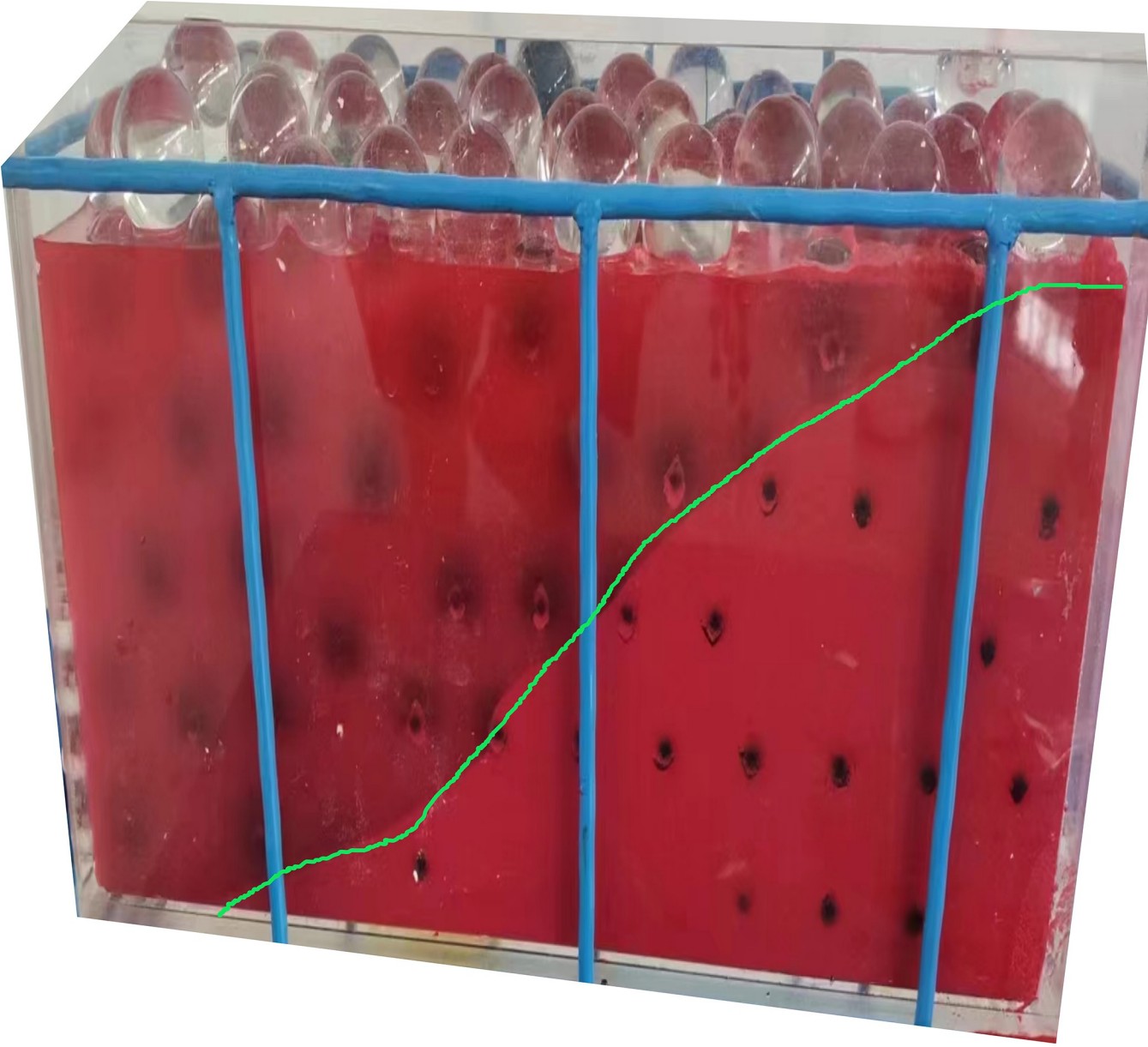

**Fig 9. Distribution of slurry deposition in the experiment.**

injection speed of 0.2m/s can make the slurry flow and spread fully inside the pore space better than that of 0.1m/s. This indicates that 0.2 m/s grouting speed can better make the slurry flow and spread inside the pore space. In addition, the flow line diagrams in XY plane and longitudinal section also proved this point, as shown in Fig 14.

### 4.3 Analysis and discussion of numerical simulation results

The above slurry flow changes in the pore space are monitored, as shown in Fig 15A–15C three monitoring lines. The three lines are at different distances from the inside of the gangue pore to the mouth of the grouting pipe, which can represent the slurry diffusion flow characteristics and diffusion law at different gangue backfill depths. The velocity along the a, b, c three lines is calculated, and its velocity change along the course is shown in Fig 15.

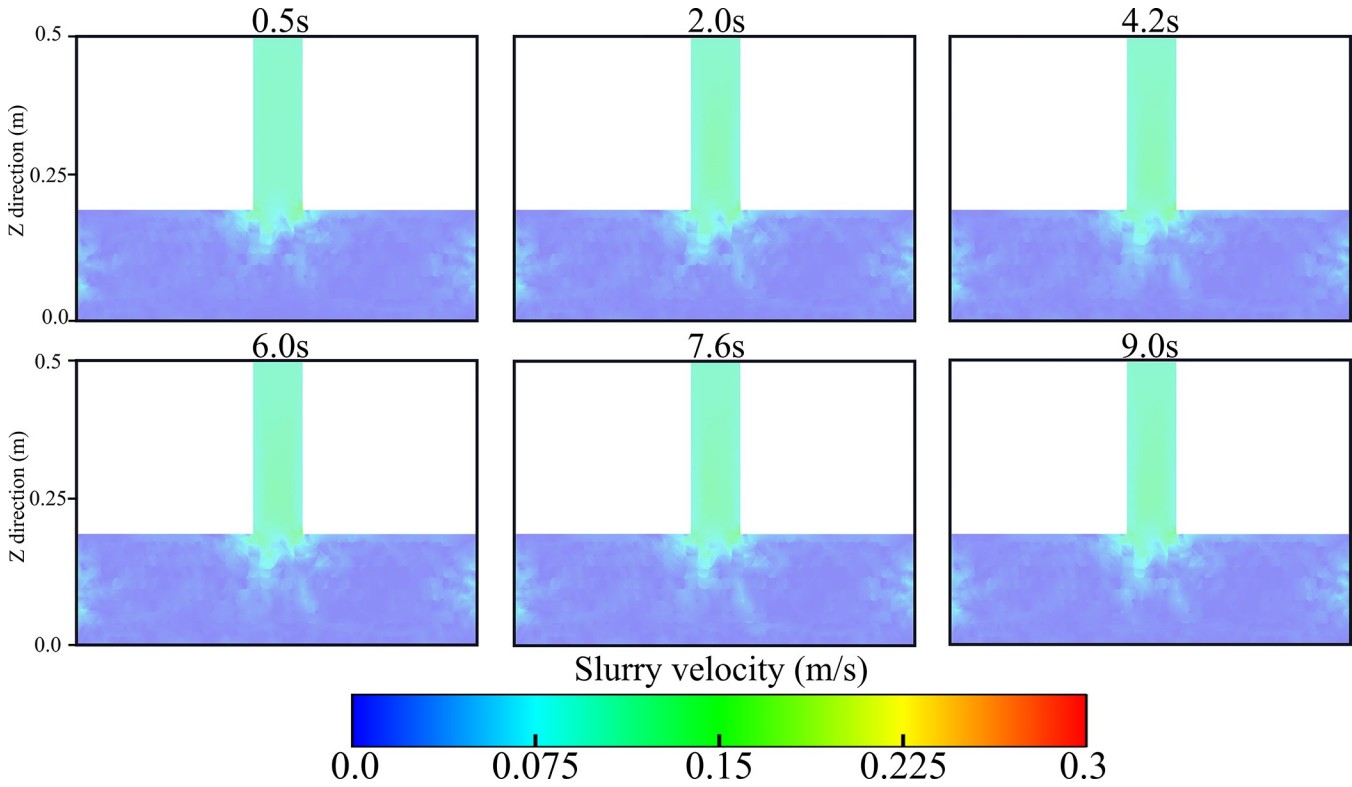

**Fig 10. Velocity cloud at different moments.**

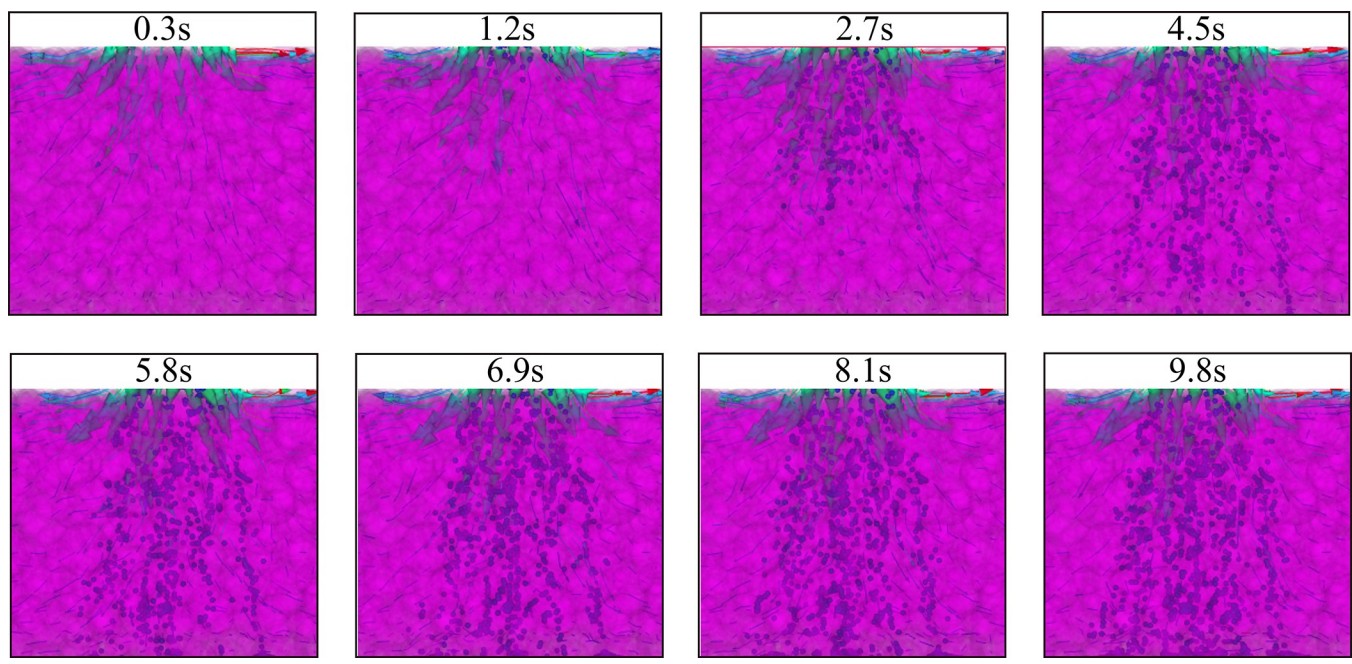

**Fig 11. Gangue internal pore flow vector map.**

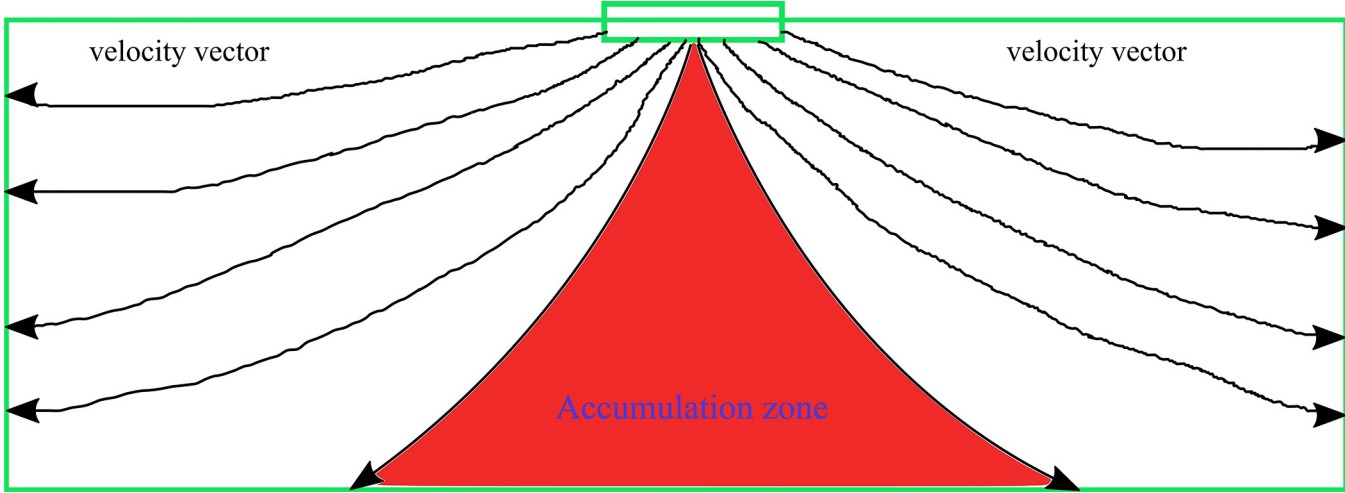

**Fig 12. Schematic flow vectors of the profile.**

Fig 15 demonstrates the variation of slurry diffusion velocity within the gangue-filled zone at different depths. Among them, the 0.35 m point is located in the center of the grouting pipe, and the slurry flow direction on both sides of this position is opposite, so the velocity direction is also opposite. In Fig 15(A), the maximum value of the slurry velocity on the monitoring line

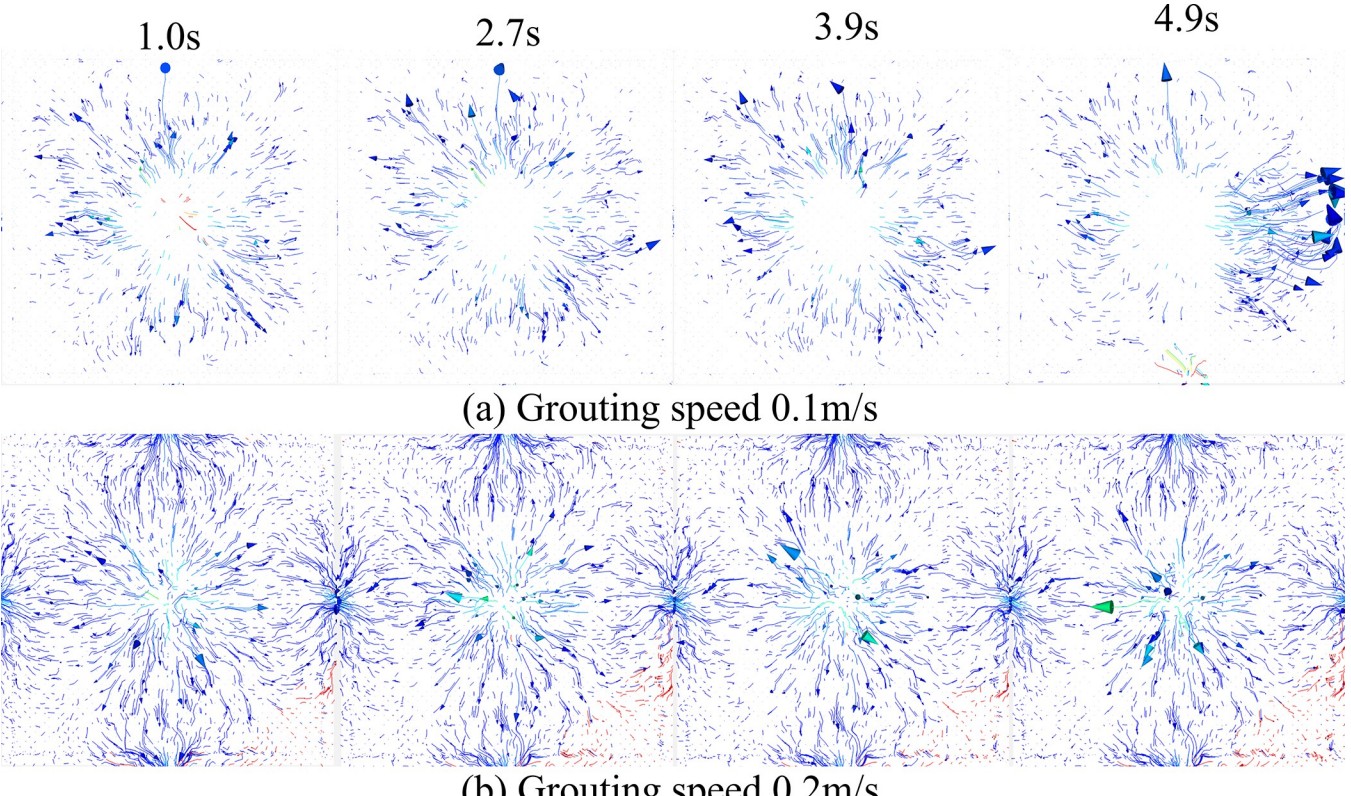

**Fig 13. Change of velocity vector in XY direction.**

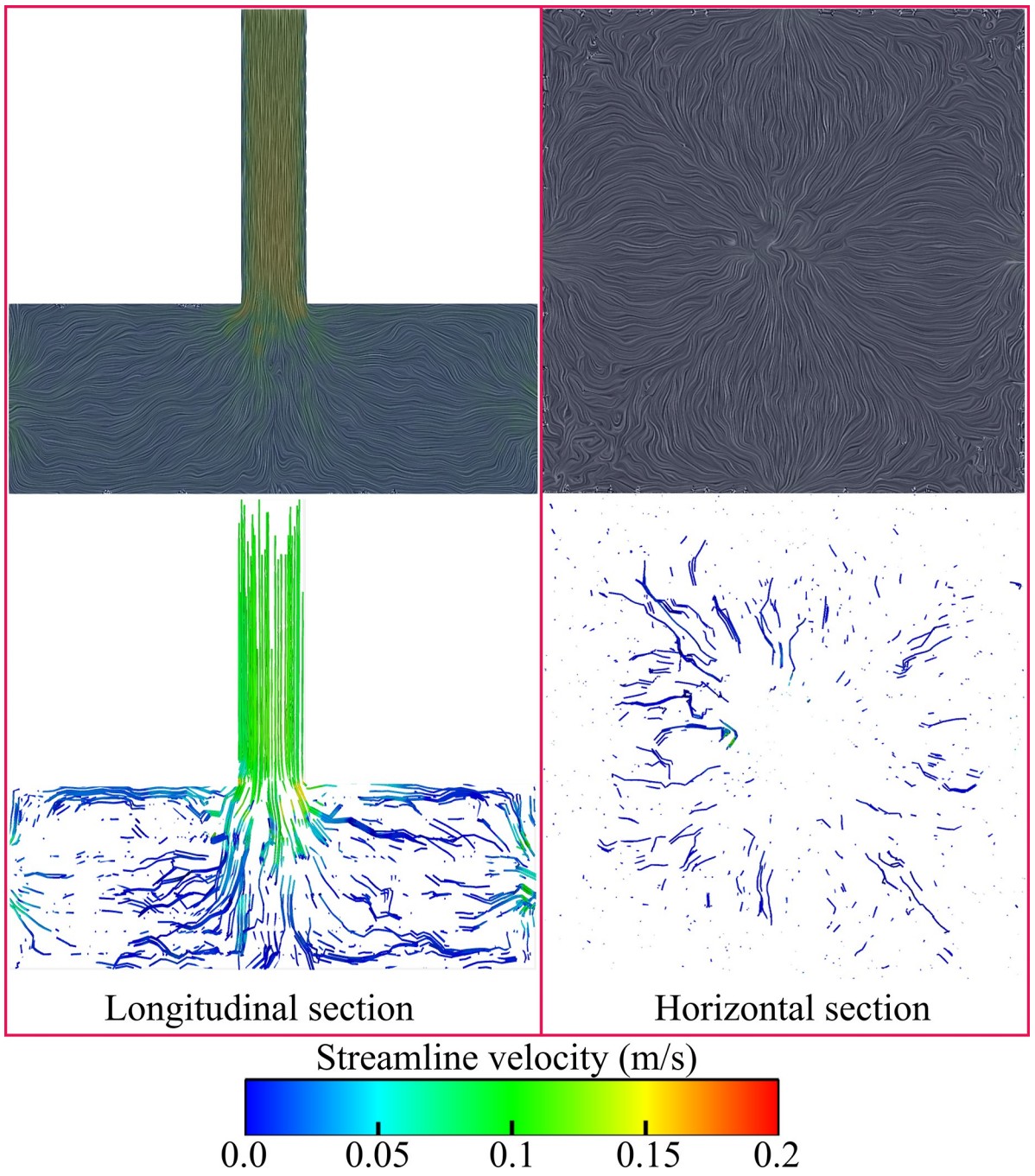

**Fig 14. The variation of the flow line in the longitudinal and transverse sections.**

on both sides of the grouting pipe is 0.065–0.1 m/s. In Fig 15(B) and 15(C), the maximum value of the slurry velocity along the monitoring line is 0.018–0.029 m/s and 0.018–0.020 m/s, respectively. furthermore, the velocity around the grouting pipe can be observed on each monitoring line to reach the phenomenon of extreme values. The slurry flow velocity gradually decreases as the distance from the grouting pipe increases, and this phenomenon is especially obvious at higher flow velocities. However, at lower flow rates, this phenomenon is not evident

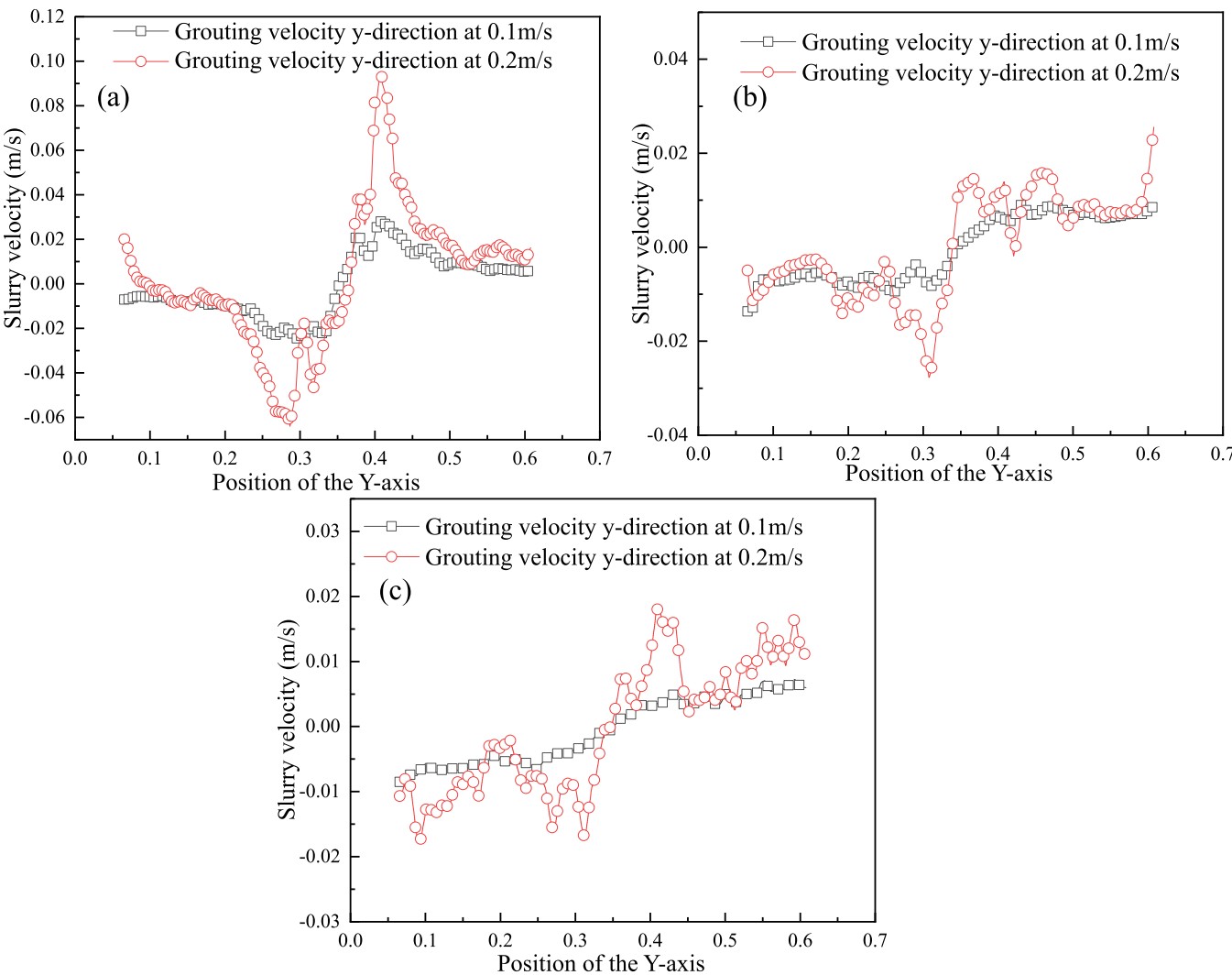

**Fig 15.** (a)-(c) Velocity variations along the three monitoring lines [32].

deeper into the gangue pore space. In addition, it is worth noting that the flow rate of slurry in the gangue-filled zone at different depths is also different. Overall, the greater the depth of slurry injection, the greater the resistance of the gangue layer to flow, resulting in a smaller steady flow rate of slurry within the pore space.

In addition, further analysis of the particles in the slurry injection pipe to the formation of this period of time in the direction of gravity velocity size changes, as shown in Fig 16 [32].

In Fig 16, it is clearly observed that there is a sharp increase in particle velocity in the time period 0–1 second. During this time, the slurry particles are mainly moving inside the pipe, so the velocity increase is caused by the effect of gravity. However, after 1 second, the particle velocity begins to decrease gradually. This decrease is caused by the pore resistance of the gangue structure, which affects the decay of the slurry velocity, thus further affecting the velocity of the particles. As the particles penetrate deeper into the gangue layer, the trend of particle velocity reduction becomes more gentle.

Overall, throughout the slurry injection into the gangue layer, the velocity of the slurry and particles in the pipe increases rapidly under the influence of gravity. However, as the slurry

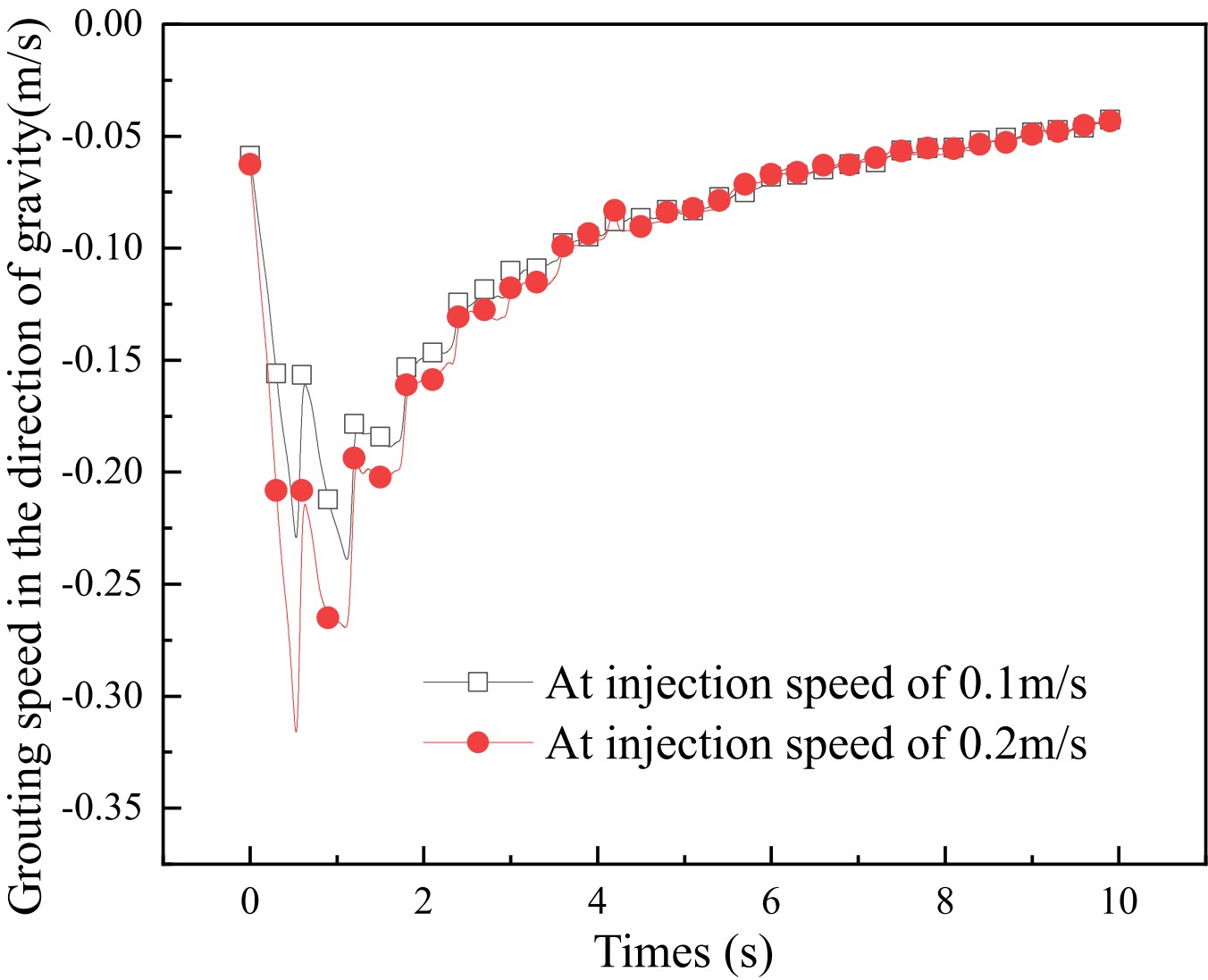

**Fig 16. Variation of slurry particle velocity with time [32].**

enters the gangue pores, the velocity of the slurry and particles decreases. This decreasing trend can be reflected by the gradual decrease of the slope of the velocity, which finally leads to the stabilization of the particle velocity.

To summarize, by observing on the XY plane axis of the gangue porous structure, it can be clearly seen that the slurry velocity and particle velocity decrease gradually with the increase of injection depth and plane diffusion range. In addition, the injection speed will also have an effect on the speed of slurry diffusion.

## 5. Conclusions

The experiment and simulation of grout diffusion and the distribution of gangue pore structure particles were studied in detail in this study, including the horizontal and longitudinal deposition of grout particles and the law of grout diffusion. The following conclusions can be drawn from this study.

(1) The existence of gangue pores will change the diffusion path of slurry. After the slurry enters the gangue pore space, the slurry flows in the pore space and conducts and diffuses along the pore gap, thus changing the slurry transfer path. This change of diffusion path will affect the flow rate and distribution of slurry, showing that the particle velocity of the grouting pipe increases rapidly under the action of slurry weight force. From the gangue pore area to the initial particle phase, the particle velocity gradually decreases, which is due to the decay of the slurry velocity caused by the gangue structure's resistance to the pore, thus affecting the particle velocity. With the increase of the depth of particles entering gangue layer, the particle velocity also increases. The decreasing trend of particle velocity is relatively smooth.

(2) During the grouting process, the particles in the gangue pores may deposit and block the pores. When the particles in the slurry enter the interior of the pore space, due to the interaction of the size and shape of the particles with the characteristics of the pore space, the particles may be deposited in the pore space and form a blockage. Such particle deposition and clogging will reduce the fluidity and permeability of the slurry.

(3) The presence of gangue pores increases slurry flow resistance. The accumulation of particles in the pore space and the limitation of pore shape will hinder the slurry flow. This pore resistance reduces the slurry flow rate and thus affects the particle velocity and distribution. In addition, the size and distribution of pore space also affect the permeability of gangue layer, that is, the ability of slurry and particles to penetrate and transport in the pore space. Specifically, the increase of injection depth and surface diffusion range will lead to a gradual decrease in slurry speed and particle speed, and the injection speed will also affect the slurry diffusion speed.

In summary, this study verifies the effect of pore injection of coal gangue on slurry velocity and particle velocity. The pore structure of gangue causes resistance to slurry flow and particle movement in the process of grouting, resulting in lower particle velocity. The research results are of great significance to the design and optimization of gangue pore grouting technology, and provide theoretical basis and practical guidance for the consolidation and stability of strata.

## Author Contributions

**Conceptualization:** Shuquan Guo, Xiaoyuan Xue.

**Data curation:** Shuquan Guo, Boqiang Wu, Xiaoyuan Xue.

**Formal analysis:** Boqiang Wu.

**Investigation:** Zhongkui Ji, Lijun Gao.

**Methodology:** Zhongkui Ji.

**Resources:** Kui Sun.

**Software:** Zhongkui Ji.

**Supervision:** Boqiang Wu.

**Validation:** Pan Chen.

**Visualization:** Lijun Gao.

**Writing – original draft:** Kui Sun, Wanchao Ma.

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
