## [Decision Letter · Decision Letter 0]

30 Oct 2023

PONE-D-23-29771Study on the Diffusion and Deposition Law of Pore Slurry in Gangue Filling Zone Based on CFD-DEM CouplingPLOS ONE

Dear Dr. wu,

Thank you for submitting your manuscript to PLOS ONE. After careful consideration, we feel that it has merit but does not fully meet PLOS ONE’s publication criteria as it currently stands. Therefore, we invite you to submit a revised version of the manuscript that addresses the points raised during the review process.

In addition to reviewers comments, the authors are suggested to also include details of time step size and percentage difference between CFD-DEM and experimental results.

We look forward to receiving your revised manuscript.

Kind regards,

Muhammad Shakaib, PhD

Academic Editor

PLOS ONE

 [YES]. 

[This work is supported by Tiandi Science and Technology Co. Ltd. Science and Technology Innovation Venture Capital Special Project (2023-TD-ZD004-003), Science and Technology Innovation Fund of Xi’an Research Institute of CCTEG (2023XAYJS11).]

 [YES]

6. PLOS requires an ORCID iD for the corresponding author in Editorial Manager on papers submitted after December 6th, 2016. Please ensure that you have an ORCID iD and that it is validated in Editorial Manager. To do this, go to ‘Update my Information’ (in the upper left-hand corner of the main menu), and click on the Fetch/Validate link next to the ORCID field. This will take you to the ORCID site and allow you to create a new iD or authenticate a pre-existing iD in Editorial Manager. Please see the following video for instructions on linking an ORCID iD to your Editorial Manager account: " ext-link-type="uri" xlink:type="simple">https://www.youtube.com/watch?v=_xcclfuvtxQ".

7. We note that Figure(s) 2, 3, 4, 7, 8, 9, 10, 12 and 13 in your submission contain copyrighted images. All PLOS content is published under the Creative Commons Attribution License (CC BY 4.0), which means that the manuscript, images, and Supporting Information files will be freely available online, and any third party is permitted to access, download, copy, distribute, and use these materials in any way, even commercially, with proper attribution. For more information, see our copyright guidelines: http://journals.plos.org/plosone/s/licenses-and-copyright.

a. You may seek permission from the original copyright holder of Figure(s) 2, 3, 4, 7, 8, 9, 10, 12 and 13  to publish the content specifically under the CC BY 4.0 license. 

8. Please upload a copy of Figure 1 and 7, to which you refer in your text. If the figure is no longer to be included as part of the submission please remove all reference to it within the text.

Reviewers' comments:

Reviewer's Responses to Questions

**Comments to the Author**

1. Is the manuscript technically sound, and do the data support the conclusions?

Reviewer #1: Yes

Reviewer #2: Yes

Reviewer #3: Partly

2. Has the statistical analysis been performed appropriately and rigorously? 

Reviewer #1: Yes

Reviewer #2: Yes

Reviewer #3: No

3. Have the authors made all data underlying the findings in their manuscript fully available?

Reviewer #1: No

Reviewer #2: No

Reviewer #3: Yes

4. Is the manuscript presented in an intelligible fashion and written in standard English?

Reviewer #1: Yes

Reviewer #2: No

Reviewer #3: No

5. Review Comments to the Author

Reviewer #1: The manuscript reports the simulation and experimental study of filling process in the gangue pore injection. The effects of various factors including slurry velocity and particle velocity are modelled. The interesting results are the particle and slurry flow dynamics induced by the grouting process. The experimental data are presented to compare with simulation results. The topic is relevant to many industrial processes, and is of importance. The manuscript is fairly clear with quite a bit of systematic results. I have following comments for the authors to consider.

Introduction: Up to line 58 on page 2, the manuscript only explains the importance of filling process. It is excessive. The introduction is a bit too general, lack of depth. What are the models in literature? What are the limitations of these models?

As for the simulation methods in this work: it is not clear how CFD and DEM are coupled. Please provide the details in section 4.1.

Grassman principle ---Grassmann principle

Equation (2) is not complete.

In table 1, gangue –Gangue

Above Figure 5 in line 285: As shown in Fig. 5. ---This is not a sentence.

Figure 5: Colors for the labels are not clear. The contrast can be improved to help visualize the particles.

Figure 8: a series of snapshots should be presented so the shell-shape should be clearly visible. At the moment, the shape is labelled by the curve. But the data do not show it clearly.

Reviewer #2: DEM Coupling: Enhancing Clarity and Simulation Details"

The paper titled "Study on the Diffusion and Deposition Law of Pore Slurry in Gangue Filling Zone Based on CFD-DEM Coupling" provides a compelling integration of CFD and experimental analysis. Although the text is well-composed, several areas require further clarification and elaboration:

1. The abstract lacks sufficient technical details and methodology. A thorough update of the abstract is essential to provide a comprehensive overview of the study.

2. Clear elucidation is required concerning the number of parameters and their specific impacts within the context of the study. Additionally, addressing uncertainties regarding the fixation of external parameters in the future, backed by relevant literature, is necessary.

3. Please include more detailed information on the DEM, such as particle size and other relevant properties.

4. Consider replacing section 2.2 with a graphical illustration to enhance understanding.

5. Elaborate on the method of particle-particle interaction employed, considering the existing lack of clarity regarding the CFD and DEM methodologies.

6. Omit ambiguous statements like "In CFD, according to the Reynolds number, flows can be classified into laminar, transitional, and turbulent flows."

7. Provide clear information regarding the Reynolds number in the simulation and the approach used to determine the turbulence model.

8. Justify the use of the DEM approach in the study, and consider discussing alternative methods, such as DPM, along with a comparative analysis of computational costs.

9. Provide a comprehensive description of the fluid properties and specifications to improve clarity.

10. Clearly articulate the simulation process for the CFD-DEM coupling, including information on the use of commercial software or custom coding.

11. Improve the clarity of Figure 4 by providing a concise explanation of its components, and ensure that the figure accurately represents the results of the CFD analysis.

12. Enhance the readability of Figure 5 and provide a technical discussion to facilitate a better understanding of the results it presents.

13. Include a detailed section on grid generation and verification, as these aspects are currently absent.

14. Address the issue with the maximum value for slurry velocity in Figure 9 by adjusting the scaling and ensuring clear visibility.

15. Improve the legibility of vectors in Figure 10, considering the 3D nature of pore flows and the necessity of representing them in a 2D format. Include quantitative values throughout the manuscript to enhance the technical analysis.

In conclusion, this manuscript necessitates textual revisions and an improved presentation of simulation details. Addressing the concerns in the text and providing a clear depiction of the simulation, particularly Figure 13, is crucial. Please consider these recommendations to ensure the clarity and accuracy of your research.

Reviewer #3: Based on my review of the work, I suggest that the authors address the following observations before it is recommended for publication in PLoS ONE. Please refer to the attached document. I strongly suggest the paper be reviewed by a nature English speaker before submitted the revised copy.

6. PLOS authors have the option to publish the peer review history of their article (what does this mean?). If published, this will include your full peer review and any attached files.

Reviewer #1: No

Reviewer #2: **Yes: **Mehrdad Mesgarpour

Reviewer #3: No

---

## [Author Response · Author response to Decision Letter 0]

8 Nov 2023

Dear Reviewers:

Thank you for your letter and for the reviewers’ comments concerning our manuscript entitled “Study on the Diffusion and Deposition Law of Pore Slurry in Gangue Filling Zone Based on CFD-DEM Coupling” (ID: PONE-D-23-29771). Those comments are all valuable and very helpful for revising and improving our paper, as well as the important guiding significance to our researches. We have studied comments carefully and have made correction which we hope meet with approval. (Changes in the text are marked in red)

Based on my review of the work, I suggest that the authors address the following observations before it is recommended for publication in PLoS ONE.

1. Abstract: Lines 13, 19, 25 and all through. When writing scientific papers, avoid using personal pronouns such as "we", "us", etc.

Thanks to your suggestion, we have revised the summary again and removed such as "we", "us", etc.

2. Introduction: It is recommended to have the work proofread by a native English speaker to ensure that there are no grammatical errors. In line 62 and throughout, insert the reference number immediately after the author's name. It is recommended to use "Gangue" for gangue, and to write the novelty of the work clearly at the end of the introduction.

Thanks for your suggestions, we have revised the whole article to modify the sentence, and changed gangue to Gangue.

3. CFD is not limited to Navier-Stokes equations. There is no doubt that the Navier–Stokes equation applies to single-phase moving fluid in Darcy-Forchheimer laminar flow regimes. Algebraic yPlus equations, Low Raynolds Average Navier - Stokes equations, Spallart Almaras equations, L-Level equations, and k-e and k-w equations apply to flows outside the laminar region for fast-flowing fluids. Kindly visit Otaru Samuel 2021 https://iopscience.iop.org/article/10.1088/2053-1591/abf3e2

Yes, your suggestion is very good. We have studied the Kindly visit Otaru Samuel 2021 article, which constructed the gaps and channels through CT 3D modeling. Due to the constructed pore channels, the mesh density and accuracy will be improved during the meshing of CFD. And grid independence verification is required in CFD. The core of CFD-DEM coupling is the relationship between particle size and mesh size. In this study, because gangue particles are fixed porous media, the force calculation is not considered, and only the motion and deposition of slurry particles are considered. In this study, a mesh larger than 10 times the volume of particles is adopted and the grid neutrality is verified. References are also cited.

The viscosity of the slurry tested in this study is 1.0e-5 m/s2 and the density is 1100 kg/m3. The parameters of grout have an important influence on the flow state in the pores, so the law of grout diffusion and particle settlement can be correctly analyzed on the basis of determining the parameters of grout and particle. On the basis of determining the material for numerical simulation, the convergence of the mesh and the relationship between the mesh particles are further analyzed, as shown in the Fig.4.

Fig. 4 Calculation grid of fluid domain and its relation with particles

In Fig. 4, when the fluid mesh decreases from 0.02m to 0.01m, the monitoring of the flow velocity at the same point shows no significant change, indicating convergence under the conditions of 0.02-~0.01m. In addition, considering whether the drag model is satisfied between the fluid mesh and particle size, The size of the skeleton particles is close to or even larger than that of the fluid grid, but because the skeleton particles only occupy the volume of the fluid without calculating the drag force, only the relationship between the size of the grout particles and the fluid grid is required. In the numerical simulation, the size of the grout particles is 3 times lower than that of the fluid grid, so it meets the requirements of coupling calculation.

4. The continuity equation is defined by Equation 2, which must equal zero.

We are very sorry for the display error caused by the formula format error. We have modified it.

5. Section 2.2 and 3 seem like reading literature. It needs to be rewritten and reorganized in third person singular. There must be clear details on how the research was conducted in this section.

Thanks for your suggestion, we have revised sections 2.2 and 3. The content of this section has been restated.

6. Explain the procedure for characterizing pore structures, the experimental procedure for analyzing the hydrodynamics of packed structures, the parameters measured, the physics solved, boundary conditions for modeling, mesh structure type, and mesh independence study.

Thanks to your suggestion, we have included in the corresponding Chapter 3 the experimental procedure for analyzing the fluid dynamics of filled structures, the packing pattern, and the grid problem solved by numerical simulation.

7. How does the tortuosity of these packed structures affect their hydrodynamics?

Thanks for your suggestion, we will add a flow diagram to illustrate the effect of pore structure on flow.

Fig. 14 The variation of the flow line in the longitudinal and transverse sections

Comments to the Author

Reviewer #1: The manuscript reports the simulation and experimental study of filling process in the gangue pore injection. The effects of various factors including slurry velocity and particle velocity are modelled. The interesting results are the particle and slurry flow dynamics inducedby the grouting process. The experimental data are presented to compare with simulation results. The topic is relevant to many industrial processes, and is of importance. The manuscript is fairly clear with quite a bit of systematic results. I have following comments for the authors toconsider.

Introduction: Up to line 58 on page 2, the manuscript only explains the importance of filling process. It is excessive. The introduction is a bit too general, lack of depth. What are the models in literature? What are the limitations of these models?

Thanks to your suggestion, we have revised the introduction to remove some descriptive paragraphs and highlight the shortcomings of the current study (in terms of pore structure and particle motion deposition law), while highlighting the characteristics of CFD-DEM coupling in terms of structural pore structure and particle flow.

As for the simulation methods in this work: it is not clear how CFD and DEM are coupled. Please provide the details in section 4.1.

Grassman principle ---Grassmann principle

Equation (2) is not complete.

Due to a problem with the editor, it is not complete. We have modified it.

In table 1, gangue –Gangue

Thank you for your suggestion, we have made changes in the table.

Above Figure 5 in line 285: As shown in Fig. 5. ---This is not a sentence.

We have modified the content to make it smooth.

Figure 5: Colors for the labels are not clear. The contrast can be improved to help visualize the particles.

We have modified Figure 5 to highlight the state of particle distribution and diffusion range.

Figure 8: a series of snapshots should be presented so the shell-shape should be clearly visible. At the moment, the shape is labelled by the curve. But the data do not show it clearly.

We are very sorry, because the slurry particles are stained, which will cause the liquid to turn the whole slurry red, so we can only pass the final stable state

Reviewer #2: DEM Coupling: Enhancing Clarity and Simulation Details"

The paper titled "Study on the Diffusion and Deposition Law of Pore Slurry in Gangue Filling Zone Based on CFD-DEM Coupling" provides a compelling integration of CFD and experimental analysis. Although the text is well-composed, several areas require further clarification and elaboration:

1. The abstract lacks sufficient technical details and methodology. A thorough update of the abstract is essential to provide a comprehensive overview of the study.

Thanks to your suggestion, we have revised the summary to condense the core conclusions and quantitative indicators.

2. Clear elucidation is required concerning the number of parameters and their specific impacts within the context of the study. Additionally, addressing uncertainties regarding the fixation of external parameters in the future, backed by relevant literature, is necessary.

Yes, we mainly compare the experimental and numerical simulation parameters to analyze the pore flow of particles in detail through numerical simulation.

3. Please include more detailed information on the DEM, such as particle size and other relevant properties.

In Table 1, we have described the particle parameters, in addition, we added gangue particles and grout particles to illustrate.

4. Consider replacing section 2.2 with a graphical illustration to enhance understanding.

Yes, thanks for your suggestion, we added the coupled calculation flowchart.

In this study, the above governing equations can be described in detail in the coupling flow through Fig. 2.

Fig. 2 Coupling flow chart of slurry in gangue flow

5. Elaborate on the method of particle-particle interaction employed, considering the existing lack of clarity regarding the CFD and DEM methodologies. 

The CFD-DEM coupling method can realize the modeling of particle skeleton structure with high precision. At the same time, the particle state of different scales can be studied by calculating the force of micro-particles through drag force. More importantly, the motion trajectory and distribution range of individual particles can be studied. This is the characteristic of this study and the important reason for adopting this method.

6. Omit ambiguous statements like "In CFD, according to the Reynolds number, flows can be classified into laminar, transitional, and turbulent flows."

Yes, thank you for your advice, We delete In the article. "In CFD, according to the Reynolds number, flows can be classified into laminar, transitional, and turbulent flows." and other related statements

7. Provide clear information regarding the Reynolds number in the simulation and the approach used to determine the turbulence model.

Thanks for your suggestion, we have added Reynolds number to the article to judge the state of flow. 

8. Justify the use of the DEM approach in the study, and consider discussing alternative methods, such as DPM, along with a comparative analysis of computational costs.

In the research, DEM method must be adopted to realize the construction of porous media, while DPM cannot realize the model of particle skeleton, so it cannot be adopted. In addition, the interaction between slurry particles and skeleton particles in the process of particle flow should be considered. We use 64 core servers, and the efficiency of particle and grid volume is good in the current situation.

9. Provide a comprehensive description of the fluid properties and specifications to improve clarity.

Thanks to your suggestion, we increase the parameters of the fluid (viscosity and density) to provide a comprehensive description of the fluid's state.

10.Clearly articulate the simulation process for the CFD-DEM coupling, including information on the use of commercial software or custom coding.

In this study, we adopted the open-source program CFDEM to realize the flow of slurry particles in the gangue pore skeleton. The detailed coupling procedure was reworked in Section 2 and the coupling flow diagram was added.

11. Improve the clarity of Figure 4 by providing a concise explanation of its components, and ensure that the figure accurately represents the results of the CFD analysis.

The porosity of the particles is consistent with the porosity taken in the test, and the porosity is calculated by the volume of the water tank occupied by the particles in the experiment. The fluctuation range of the pore distribution of each section monitored in the numerical simulation is close to that of the experiment, so the numerical simulation can meet the requirements.

12. Enhance the readability of Figure 5 and provide a technical discussion to facilitate a better understanding of the results it presents.

Yes, we have modified the clarity of Figure 5 in the paper, which can clearly show the distribution and deposition range of particles under different grouting speeds.

13. Include a detailed section on grid generation and verification, as these aspects are currently absent.

In Section 3, we add a convergence evaluation of mesh and particle scales to demonstrate the effectiveness and convergence of the numerical simulation.

14. Address the issue with the maximum value for slurry velocity in Figure 9 by adjusting the scaling and ensuring clear visibility.

Thanks to your suggestions, we have modified Figure 9 to highlight the range of clarity and maximum values.

15. Improve the legibility of vectors in Figure 10, considering the 3D nature of pore flows and the necessity of representing them in a 2D format. Include quantitative values throughout the manuscript to enhance the technical analysis.

Yes, we have modified Figure 10 to make it clearer to see the flow of vectors through the pores.

In conclusion, this manuscript necessitates textual revisions and an improved presentation of simulation details. Addressing the concerns in the text and providing a clear depiction of the simulation, particularly Figure 13, is crucial. Please consider these recommendations to ensure the clarityand accuracy of your research.

Let's modify Figure 13 to make it fit.

Reviewer #3: Based on my review of the work, I suggest that the authors address the following observations before it is recommended for publication in PLoS ONE. Please refer to the attached document. I strongly suggest the paper be reviewed by a nature English speaker before submitted therevised copy.

We have improved the quality of the full text and polished it.

---

## [Decision Letter · Decision Letter 1]

29 Nov 2023

PONE-D-23-29771R1Study on the Diffusion and Deposition Law of Pore Slurry in Gangue Filling Zone Based on CFD-DEM CouplingPLOS ONE

Dear Dr. wu,

Thank you for submitting your manuscript to PLOS ONE. After careful consideration, we feel that it has merit but does not fully meet PLOS ONE’s publication criteria as it currently stands. Therefore, we invite you to submit a revised version of the manuscript that addresses the points raised during the review process.

The authors are suggested to improve the Figures and related discussions and make grammatical changes as recommended by the reviewers.

We look forward to receiving your revised manuscript.

Kind regards,

Muhammad Shakaib, PhD

Academic Editor

PLOS ONE

Journal Requirements:

Reviewers' comments:

Reviewer's Responses to Questions

**Comments to the Author**

1. If the authors have adequately addressed your comments raised in a previous round of review and you feel that this manuscript is now acceptable for publication, you may indicate that here to bypass the “Comments to the Author” section, enter your conflict of interest statement in the “Confidential to Editor” section, and submit your "Accept" recommendation.

Reviewer #2: All comments have been addressed

Reviewer #3: (No Response)

2. Is the manuscript technically sound, and do the data support the conclusions?

Reviewer #2: No

Reviewer #3: Yes

3. Has the statistical analysis been performed appropriately and rigorously? 

Reviewer #2: No

Reviewer #3: Yes

4. Have the authors made all data underlying the findings in their manuscript fully available?

Reviewer #2: Yes

Reviewer #3: Yes

5. Is the manuscript presented in an intelligible fashion and written in standard English?

Reviewer #2: Yes

Reviewer #3: Yes

6. Review Comments to the Author

Reviewer #2: Thank you for the revised version of the manuscript. While efforts to enhance quality and address previous concerns are evident, several key issues still persist. My feedback on the current version is as follows:

Figure 1's Effectiveness: I am uncertain about the usefulness of Figure 1 in aiding the reader's understanding. The absence of scale, color details, and dimensions limits its effectiveness. I suggest replacing it with a more informative figure.

Enhancements for Figure 2: I appreciate Figure 2, but it would benefit from additional details and descriptions to better convey its intended message.

Concerns about Figure 3: The lack of scale in Figure 3 is problematic, as it features an image of an unknown material. This omission is surprising and diminishes the figure's clarity. Also, the relationship between Figure 3a and 3b is unclear and needs explanation.

Clarification Needed for Figure 4: The purpose of Figure 4 is not apparent. It raises several questions, particularly regarding the dark points in the figure. Despite the minimum particle size being 0.008m, the grid size is noted as 0.01m. Can you clarify what is meant by 'satisfying the relationship' in this context?

Persistent Concerns: Unfortunately, most of the concerns I raised in my last review remain unaddressed in this version of the manuscript.

Reviewer #3: Most of my comments/observations have been addressed by the authors, but the work still needs to be corrected before it can be published. In my earlier submission, I emphasized that authors should avoid using personal pronouns in their research writing. This area is still not fully addressed. In the Abstract section, for example, "In the plane flow field, we observe"; remove "we" and re-write it as..."In the plane flow field, it was observed that......". Check all through the work as well.

7. PLOS authors have the option to publish the peer review history of their article (what does this mean?). If published, this will include your full peer review and any attached files.

Reviewer #2: **Yes: **Mehrdad Mesgarpour

Reviewer #3: No

---

## [Author Response · Author response to Decision Letter 1]

2 Dec 2023

Dear Reviewers:

Thank you for your letter and for the reviewers’ comments concerning our manuscript entitled “Study on the Diffusion and Deposition Law of Pore Slurry in Gangue Filling Zone Based on CFD-DEM Coupling” (ID: PONE-D-23-29771). Those comments are all valuable and very helpful for revising and improving our paper, as well as the important guiding significance to our researches. We have studied comments carefully and have made correction which we hope meet with approval. (Changes in the text are marked in red)

Journal Requirements:

Please review your reference list to ensure that it is complete and correct. If you have cited papers that have been retracted, please include the rationale for doing so in the manuscript text, or remove these references and replace them with relevant current references. Any changes to thereference list should be mentioned in the rebuttal letter that accompanies your revised manuscript. If you need to cite a retracted article, indicate the article’s retracted status in the References list and also include a citation and full reference for the retraction notice.

Thank you for your suggestion. As the reviewer suggested deleting some descriptive statements, some references supported the descriptive statements. Therefore, after deleting these related statements, we cannot still place the references in that position. Therefore, we have removed the relevant excess references.

Reviewer #2: Thank you for the revised version of the manuscript. While efforts to enhance quality and address previous concerns are evident, several key issues still persist. My feedback on the current version is as follows:

Figure 1's Effectiveness: I am uncertain about the usefulness of Figure 1 in aiding the reader's understanding. The absence of scale, color details, and dimensions limits its effectiveness. I suggest replacing it with a more informative figure.

We have adopted a different version of Figure 1 with added scales. However, due to system issues, we uploaded it in the form of an attachment. The gangue skeleton model in Figure 1 mainly displays the pore structure filled with particles. 

Figure 1 Gangue skeleton model

Enhancements for Figure 2: I appreciate Figure 2, but it would benefit from additional details and descriptions to better convey its intended message.

Thank you for your suggestion. In the previous revision, other reviewers suggested using a computational flowchart instead of a statement description. Therefore, we have added a computational flowchart. We will add a brief statement description on top of the flowchart.

The entire calculation process can be expressed as setting the initial velocity of the fluid in CFD simulation, solving the Navier Stokes equation, and using CFD method to simulate fluid motion. Construct the pore structure of gangue particles in DEM simulation. Coupling CFD results with DEM results. At each time step, the drag force received by particles is calculated based on the fluid velocity field to update their motion state. Iterative loop until the predetermined calculation time or convergence standard is reached.

Concerns about Figure 3: The lack of scale in Figure 3 is problematic, as it features an image of an unknown material. This omission is surprising and diminishes the figure's clarity. Also, the relationship between Figure 3a and 3b is unclear and needs explanation.

Fig.3 Measurement of the stacking angle of slurry particles

Thank you for your suggestion. We have added an angle diagram in Figure 3 to illustrate that the numerical simulation and experimental results are consistent. Figure 3 shows the parameter calibration of the experimental and numerical simulation of the stacking angle of rest. When the angle of rest obtained in the experiment is consistent with the simulation results, the parameter calibration is completed. We have tested and found that α Related to β It is consistent, indicating that the parameters of the numerical simulation have been validated.

Clarification Needed for Figure 4: The purpose of Figure 4 is not apparent. It raises several questions, particularly regarding the dark points in the figure. Despite the minimum particle size being 0.008m, the grid size is noted as 0.01m. Can you clarify what is meant by 'satisfying therelationship' in this context?

Fig. 4 Calculation grid of fluid domain and its relation with particles

Thank you for your suggestion. All of our theories are based on the CFD-DEM coupling theory. In unresolved CFD-DEM coupling, the volume of moving particles should be less than 10 times that of the fluid mesh to ensure the effectiveness of the drag force during calculation. Therefore, Figure 4 shows the relationship between moving particles and skeleton particles in the process of grid independence verification. The mesh we used meets the requirements of grid independence, And it conforms to the grid size and particle size relationship of drag force in CFD-DEM coupling calculation, which can be reflected in Figure 4.

Reviewer #3: Most of my comments/observations have been addressed by the authors, but the work still needs to be corrected before it can be published. In my earlier submission, I emphasized that authors should avoid using personal pronouns in their research writing. This area is still notfully addressed. In the Abstract section, for example, "In the plane flow field, we observe"; remove "we" and re-write it as..."In the plane flow field, it was observed that......". Check all through the work as well.

Thanks to your suggestions, we have revised the language issues mentioned in the article, especially the person. All changes have been highlighted in red.

---

## [Decision Letter · Decision Letter 2]

18 Dec 2023

PONE-D-23-29771R2Study on the Diffusion and Deposition Law of Pore Slurry in Gangue Filling Zone Based on CFD-DEM CouplingPLOS ONE

Dear Dr. wu,

Thank you for submitting your manuscript to PLOS ONE. After careful consideration, we feel that it has merit but does not fully meet PLOS ONE’s publication criteria as it currently stands. Therefore, we invite you to submit a revised version of the manuscript that addresses the points raised during the review process.

We look forward to receiving your revised manuscript.

Kind regards,

Muhammad Shakaib, PhD

Academic Editor

PLOS ONE

Journal Requirements:

**Additional Editor Comments:**

The authors are suggested to provide further details of the angle of repose i.e.  importance of this angle in their study and its numerical value in simulations and experiments (α and β). Also explain how various parameters (Poisson's ratio, Young Modulus and other coefficients) in Table 1 were determined through angle of repose test.

Reviewers' comments:

Reviewer's Responses to Questions

**Comments to the Author**

1. If the authors have adequately addressed your comments raised in a previous round of review and you feel that this manuscript is now acceptable for publication, you may indicate that here to bypass the “Comments to the Author” section, enter your conflict of interest statement in the “Confidential to Editor” section, and submit your "Accept" recommendation.

Reviewer #2: (No Response)

Reviewer #3: All comments have been addressed

2. Is the manuscript technically sound, and do the data support the conclusions?

Reviewer #2: No

Reviewer #3: Yes

3. Has the statistical analysis been performed appropriately and rigorously? 

Reviewer #2: No

Reviewer #3: Yes

4. Have the authors made all data underlying the findings in their manuscript fully available?

Reviewer #2: Yes

Reviewer #3: Yes

5. Is the manuscript presented in an intelligible fashion and written in standard English?

Reviewer #2: Yes

Reviewer #3: Yes

6. Review Comments to the Author

Reviewer #2: After thorough review in three iterations, I must regrettably maintain my recommendation to reject this paper. Despite the authors' commendable efforts, which I genuinely acknowledge, the paper fails to meet the rigorous standards required for publication in PLOS ONE. My primary concern lies with the clarity and coherence of the Computational Fluid Dynamics (CFD) details and the numerical approach. While the title, content, and experimental approach are novel and understandable, the numerical modeling and description of CFD, including figures and discussions, exhibit significant discrepancies compared to the rest of the paper. This gap is not just a matter of style but of substance, undermining the paper's scientific rigour.

For instance, Figure 3 lacks crucial information, such as the range of angles being studied, leaving the reader unable to fully grasp the study's implications. This is representative of a broader issue where key aspects of the numerical analysis remain opaque, even after two rounds of revision. My previous concerns, articulated in earlier reviews, persist unaddressed. These issues are not minor and require fundamental revision, not just superficial edits. In its current form, the paper does not align with the high standards of clarity and completeness that PLOS ONE upholds. Therefore, I strongly believe that, without significant improvements in these areas, this paper should not be considered for publication in its current state.

Reviewer #3: The authors have incorporated all my suggestions into their revised submission. As an author in the field of transportation in porous media, CFD modelling, simulations, and machine learning and AI applications, I believe this topic will be of interest to audiences around porous materials, structures, and characterizations.

7. PLOS authors have the option to publish the peer review history of their article (what does this mean?). If published, this will include your full peer review and any attached files.

Reviewer #2: No

Reviewer #3: **Yes: **Dr. Abdulrazak Jinadu Otaru

---

## [Author Response · Author response to Decision Letter 2]

22 Dec 2023

Dear Editor:

Thank you for your letter and for the reviewers’ comments concerning our manuscript entitled “Study on the Diffusion and Deposition Law of Pore Slurry in Gangue Filling Zone Based on CFD-DEM Coupling” (ID: PONE-D-23-29771). Those comments are all valuable and very helpful for revising and improving our paper, as well as the important guiding significance to our researches. We have studied comments carefully and have made correction which we hope meet with approval. (Changes in the text are marked in red)

Journal Requirements:

Respond：Thanks for your suggestion, we have revised the reference. Based on the revised draft, we have placed the reference removed before in the corresponding place, and added our research value as the reference [32], which has been fully disclosed, in addition to adding a reference [7].

[32] wu, Boqiang (2023), “Slurry velocity and particle velocity”, Mendeley Data, V1, doi: 10.17632/6z2wfrw4sm.1

[7] A.J. Otaru, M.B. Samuel, Pore-level CFD investigation of velocity and pressure dispositions in microcellular structures, Materials Research Express. 8 (2021). https://doi.org/10.1088/2053-1591/abf3e2.

Additional Editor Comments:

The authors are suggested to provide further details of the angle of repose i.e. importance of this angle in their study and its numerical value in simulations and experiments (α and β). Also explain how various parameters (Poisson's ratio, Young Modulus and other coefficients) in Table 1 were determined through angle of repose test.

Respond：Thank you for your suggestion. We will provide the following answer to this question. In this study, a combination of numerical simulation and experimental methods was used to test the angle of rest. The density and size of particles can be obtained in the experiment, but it is difficult to obtain the Poisson's ratio, friction coefficient, and recovery coefficient of particles. Therefore, a static angle test is adopted, which measures the filling angle when the particles are filled, and then conducts multiple numerical simulations, changing the parameters of the particles in the numerical simulations to measure the filling static angle. When the static angle of the packing is close to the static angle of the packing, it can be considered that the parameter calibration is completed. This method is commonly used in parameter calibration of discrete element simulation.

---

## [Editor Report · Decision Letter 3]

29 Dec 2023

Study on the Diffusion and Deposition Law of Pore Slurry in Gangue Filling Zone Based on CFD-DEM Coupling

PONE-D-23-29771R3

Dear Dr. wu,

We’re pleased to inform you that your manuscript has been judged scientifically suitable for publication and will be formally accepted for publication once it meets all outstanding technical requirements.

Kind regards,

Muhammad Shakaib, PhD

Academic Editor

PLOS ONE
---

## [Editor Report · Acceptance letter]

21 Feb 2024

PONE-D-23-29771R3 

PLOS ONE

Dear Dr. Wu, 

I'm pleased to inform you that your manuscript has been deemed suitable for publication in PLOS ONE. Congratulations! Your manuscript is now being handed over to our production team.

Kind regards, 

on behalf of

Dr. Muhammad Shakaib 

Academic Editor

PLOS ONE